# Robust links in photoactive covalent organic frameworks enable effective photocatalytic reactions under harsh conditions

Jia-Rui Wang[1], Kepeng Song [1], Tian-Xiang Luan[1], Ke Cheng[1], Qiurong Wang[1], Yue Wang[1], William W. Yu [1], Pei-Zhou Li [1,2] ✉ & Yanli Zhao [2] ✉

Developing heterogeneous photocatalysts for the applications in harsh conditions is of high importance but challenging. Herein, by converting the imine linkages into quinoline groups of triphenylamine incorporated covalent organic frameworks (COFs), two photosensitive COFs, namely TFPA-TAPT-COF-Q and TFPA-TPB-COF-Q, are successfully constructed. The obtained quinoline-linked COFs display improved stability and photocatalytic activity, making them suitable photocatalysts for photocatalytic reactions under harsh conditions, as verified by the recyclable photocatalytic reactions of organic acid involving oxidative decarboxylation and organic base involving benzylamine coupling. Under strong oxidative condition, the quinoline-linked COFs show a high efficiency up to 11831.6 μmol·g$^{-1}$·h$^{-1}$ and a long-term recyclable usability for photocatalytic production of H$_2$O$_2$, while the pristine imine-linked COFs are less catalytically active and easily decomposed in these harsh conditions. The results demonstrate that enhancing the linkage robustness of photoactive COFs is a promising strategy to construct heterogeneous catalysts for photocatalytic reactions under harsh conditions.

The aerobic oxidation is a fundamental reaction in chemical science and industrial applications[1,2]. In photochemistry, reactive oxygen species (ROS) can be catalytically generated under light irradiation, which usually makes the reaction very effective and environmental-friendly. Therefore, photocatalytic approach becomes a promising way in aerobic oxidation. So far, tremendous efforts have been dedicated and various photocatalysts such as transition metal-based composites[3,4], polyoxometalates[5,6] and coordination complexes[7–9] have been synthesized for photocatalytic aerobic oxidations. Nevertheless, developing effective photocatalysts for carrying out aerobic oxidations especially in harsh conditions is of high importance but challenging.

Covalent organic frameworks (COFs) are as a group of functional porous materials in wide applications[10–14]. COFs are a group of crystalline porous organic polymers composed of organic components periodically linked by organic covalent bonds, which endowed them with chemical tunability and functionality by decorating either the organic moieties or the covalent linkages. By taking organic compounds with photosensitive properties as organic modules, photoactive COFs usually can be obtained, which exhibit a bright prospect in heterogeneous photocatalysis[15]. For instance, diverse photo-responsive COFs incorporating photosensitive pyrene[16], tetraphenylethene[17], phthalocyanine[18] have been developed for heterogeneous photocatalytic reactions. Decorating triphenylamine[19–21] and porphyrin groups[22] into the skeletons of COFs, effective heterogeneous photocatalysts were also achieved by our group for photocatalytic generation of ROS. However, during our investigations for COF-based photocatalytic aerobic oxidations, obstacles of their

[1]School of Chemistry and Chemical Engineering, Shandong Provincial Key Laboratory for Science of Material Creation and Energy Conversion, Science Center for Material Creation and Energy Conversion, Shandong University, No. 27 Shanda South Road, Ji'nan 250100, PR China. [2]School of Chemistry, Chemical Engineering and Biotechnology, Nanyang Technological University, 21 Nanyang Link, 637371 Singapore, Singapore. ✉ e-mail: pzli@sdu.edu.cn; zhaoyanli@ntu.edu.sg

stability and reusability especially in harsh conditions such as acidic, alkaline, and even strongly oxidizing conditions were always encountered. Therefore, construction of robust photo-responsive COFs is a real need to achieve for photocatalytic aerobic oxidations in harsh conditions.

In order to enhance the robustness of COFs, besides the in-situ generation of highly stable linkages such as triazine[23], pyrazine[24], imidazole groups[25] and sp[2] carbon bonds[26,27], another strategy is modifying the linkage bonds to increase the stability of constructed COFs. Although various persuasive strategies have been employed to enhance the linkage robustness of COFs[28–30], enhancing the bond stabilities of COFs for heterogeneous photocatalysis in harsh environments has not been well systematically studied. Herein, taking triphenylamine (TPA)-containing organic module[31,32] as the photosensitive building block, we report the construction of two robust COFs (Fig. 1), i.e., TFPA-TAPT-COF-Q and TFPA-TPB-COF-Q, obtained by modifying the imine linkages for photocatalytic aerobic oxidations in harsh conditions. Studies revealed that after converting the imine linkages into quinoline groups in the COFs, their stabilities are remarkably improved when their crystallinity is still well maintained. Moreover, the photochemical properties of the constructed quinoline-linked COFs exhibit a high enhancement as compared with the pristine imine-linked COFs. Catalytic investigations revealed that the synthesized COFs can effectively catalyze the generation of superoxide radical anion ($O_2^{\bullet-}$) for recyclable photocatalytic aerobic oxidation reactions under harsh conditions such as the organic acid involving oxidative decarboxylation, the organic base involving benzylamine coupling and even the reaction under strong oxidative condition for photocatalytic production of $H_2O_2$, while the pristine imine-linked COFs are easily decomposed under the same conditions for the long-term continuous reactions. Moreover, the quinoline-linked COFs display a much high efficiency even up to 11831.6 $\mu mol \cdot g^{-1} \cdot h^{-1}$ and long-term recyclable usability for photocatalytic production of $H_2O_2$, which is the highest efficiency as we know among reported COF-based photocatalysts so far for the photocatalytic production of $H_2O_2$. This study demonstrates that enhancing the linkage robustness of photoactive COFs should be a promising strategy to construct heterogeneous photocatalysts for catalytic reactions under harsh conditions.

## Results

Among the COF constructions, the polymeric condensation of aldehyde- and amine-based organic modules to give the formation of imine-linked COFs is a typical reaction[30,33,34]. Due to the reversibility of in-situ formed Schiff base imine linkages, highly crystalline COFs can be obtained. Because of the same reason, the stability of the Schiff base linked COFs is limited especially in harsh conditions. In order to

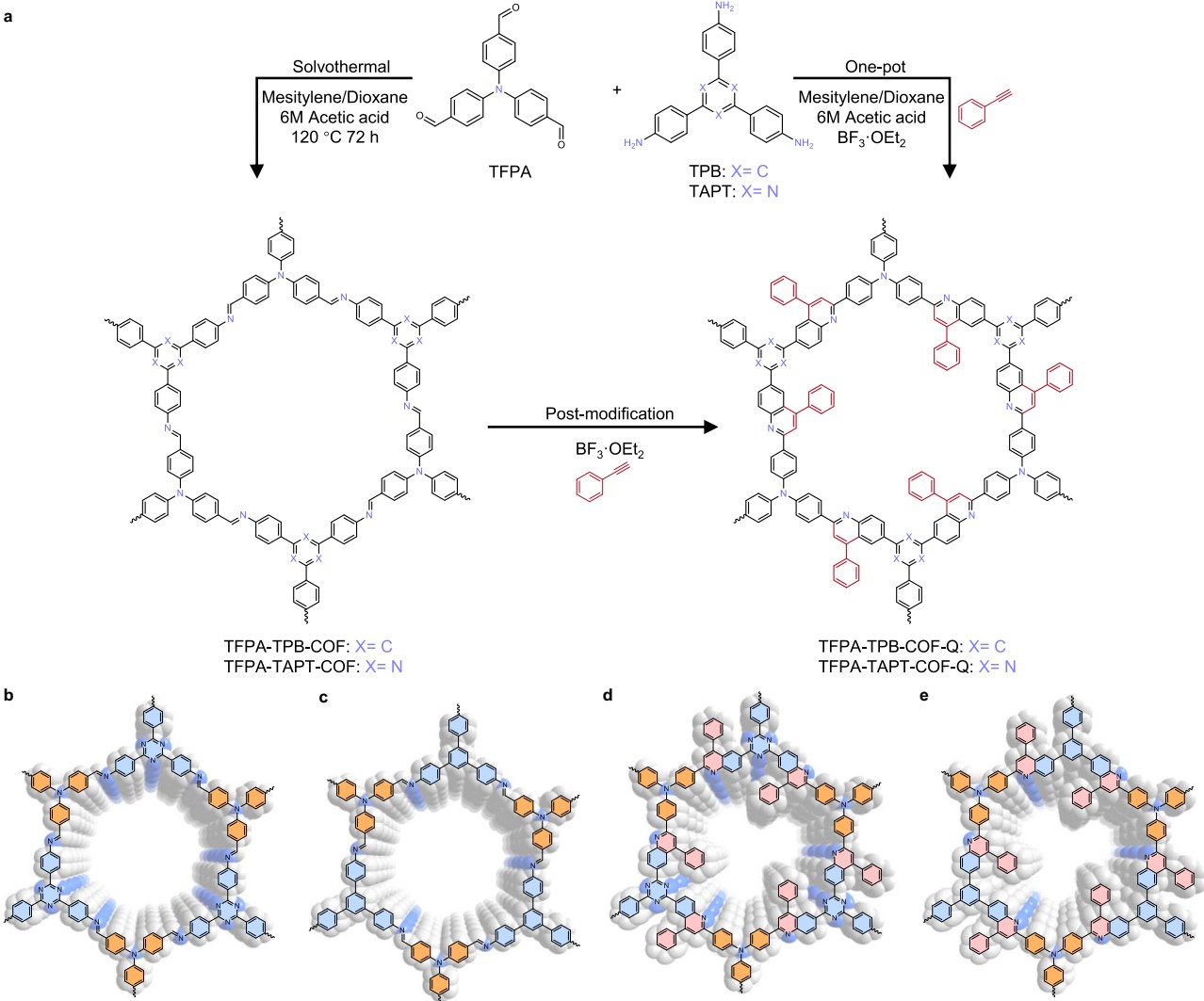

**Fig. 1 | Synthesis and structures. a** Syntheses of triphenylamine-incorporated COFs and illustrations for **b** TFPA-TAPT-COF, **c** TFPA-TPB-COF, **d** TFPA-TAPT-COF-Q, and **e** TFPA-TPB-COF-Q.

improve the bond stability of the Schiff base linkages, beside post-modification by oxidation[35] and reduction[36] as well as addition methods[37], some new approaches have recently been reported. For example, the imine bonds could be converted to substituted quinoline bonds via the Povarov reaction[38–45], to 4-carboxy-quinoline bonds via the Doebner reaction[46–48] and to nonsubstituted quinoline bonds with the rhodium-catalyzed [4 + 2] annulation[49–51]. These studies successfully demonstrated that converting the imine bonds into quinoline bonds to enhance the bond stabilities is useful for their applications in adsorption, electrocatalysis, nanofiltration and so on (Supplementary Table 1)[38–51]. Nevertheless, there are only a few examples of systematical studies on photocatalytic reactions under harsh conditions based on quinoline-linked COFs.

Triphenylamine-based organic compounds are a kind of commonly used photosensitive module in photoactive materials fabrications. To construct photoactive COFs for photocatalytic aerobic oxidations, a TPA-based organic aldehyde, tris(4-formylphenyl)amine (TFPA), was selected as the basic organic building block. Triazine units[52] as well-known electron-deficient building blocks show nitrogen-rich and planar features, which are appropriate for enhancing the process of π-electron communication. Therefore, organic amine 1,3,5-tris-(4-aminophenyl)triazine (TAPT) was then selected as electron-acceptor module for reacting with the electron-donor TFPA unit in the construction of COFs (Fig. 1a). As a typical reaction in converting imine bonds to quinoline bonds, the Povarov reaction not only converts the imine linkages into quinoline ones to enhance the bond robustness but also increases the conjugation systems by converting the original donor-acceptor system to donor-π-acceptor system, which can extend the photo-absorption capability and finally improve their performance in photocatalysis[38–45]. Moreover, the Povarov reaction can introduce benzene rings as the side groups to enhance the interactions between two neighbor layers via the longitudinal π-π stacking effect to give the generation of a highly crystalline framework. Considering these benefits it possesses, we thus chose the Povarov reaction as the method to enhance the bond robustness. After the one-pot Povarov reaction with the presence of phenylacetylene, the substituted quinoline-linkage COF, TFPA-TAPT-COF-Q, was synthesized in the presence of $BF_3 \cdot OEt_2$ in the mixture solutions of 1,4-dioxane and mesitylene (Fig. 1a, and also see synthetic details in the Supplementary Information). As a control experiment, the imine-linked COF named TFPA-TAPT-COF was also prepared without the presence of phenylacetylene according to the reported literature[53,54]. Meanwhile, the post-modification method by reacting the imine bonds in TFPA-TAPT-COF with phenylacetylene to convert it into quinoline bonds was also employed, and the obtained production was named TFPA-TAPT-COF-Q′.

In order to verify the universality of the method to convert the imine bonds into quinoline bonds to enhance the bond stability of photoactive COFs for heterogeneous photocatalysis in harsh environments, 1,3,5-tris(4-aminophenyl) benzene (TPB), as a very similar structural module with TAPT in size and symmetry, was also selected as a module, and TFPA-TPB-COF-Q and TFPA-TPB-COF-Q′ as well as their control sample TFPA-TPB-COF were synthesized by replacing TAPT with TPB to react with TFPA under the aforementioned processes (Fig. 1b–e).

Powder X-ray diffraction (PXRD) analyses were employed to elucidate the structure of the synthesized COFs (Fig. 2). In the case of TFPA-TAPT-COF-Q, the PXRD pattern is quite similar with that of TFPA-TAPT-COF-Q′, while their main peaks are different from the peaks of TFPA-TAPT-COF (Fig. 2a), indicating the crystal structures of the samples obtained from both methods are the same with each other but different from the pristine imine-linked COFs. It should be mentioned that the yield of TFPA-TAPT-COF-Q synthesized by the one-pot method is much higher than that of TFPA-TAPT-COF-Q′ synthesized by the post-modification method. Moreover, TFPA-TAPT-COF-Q could be easily scaled up to the gram level by the one-pot method (see

"Method" section). Therefore, TFPA-TAPT-COF-Q synthesized by the one-pot method is used for the following structural analyses and catalytic performance characterizations. Based on the structure of imine-linked TFPA-TAPT-COF, a possible quinoline-linked two-dimensional (2D) framework[53,54] in [3 × 3 × 3] unit with vertical AA stacking model was built using the software Materials Studio[55] to explore its crystalline structure. In the PXRD pattern of TFPA-TAPT-COF-Q, there was an intense reflection peak at $2\theta = 4.48°$, ascribed to the reflection of the (100) plane in the simulated crystal structure (Fig. 2b). Along with the main peak at 4.48°, there were also three weak peaks at 7.74, 8.92, and 11.82°, which can be attributed to the reflections of (110), (200) and (210) planes, respectively (Fig. 2b). After structural optimization by Pawley refinements, a unit cell in $P_3$ space group with the parameters of $a = b = 22.70$ Å, $c = 3.94$ Å was obtained. The fitting refinement and the experimental results showed a great resemblance, proving the reliability of the quinoline-linked 2D framework and its packed crystal structure model (Fig. 2b, c), while remarkable differences were observed between the experimental PXRD pattern and the simulated PXRD pattern from the crystal structure packed from staggered AB stacking of the quinoline-linked 2D framework (Fig. 2a).

For TFPA-TPB-COF-Q and TFPA-TPB-COF-Q′, their PXRD patterns are also matched very well with each other, while they are different with that of the imine-linked TFPA-TPB-COF (Fig. 2d), indicating that both of the one-pot method and the post-modification method can modify the imine linkages. Following the same way as that of TFPA-TAPT-COF-Q, the structure of TFPA-TPB-COF-Q was also simulated and refined. According to its reflection pattern of a sharp peak at $2\theta = 4.53°$ and three weak peaks at 7.78, 8.96, and 11.86° (Fig. 2e), a structural with vertical AA stacking of the quinoline-linked 2D framework having unit cell parameters of $a = b = 22.01$ Å, $c = 3.97$ Å was established (Fig. 2e, f)[21]. It is worth mentioning that, in their PXRD patterns, M-shaped diffraction peaks in both imine-linked COFs at the low angle usually caused by a stepwise slide of the 2D frameworks in their vertical direction disappeared[56–58], while single sharp peaks were instead observed at the related $2\theta$ areas in the two quinoline-linked COFs (Fig. 2a, d). Therefore, it was speculated that the quinoline linkage as well as the introduced side benzene ring can enhance their longitudinal π-π stacking interactions and the crystallinity of the formed COFs.

In order to study the crystalline morphology of the two quinoline-linked COFs, after ultrasonic treatment, scanning electron microscope (SEM) and transmission electron microscope (TEM) measurements were employed to observe their microscopic topography directly. As those of the related imine-linked COFs, their SEM and TEM images revealed that TFPA-TAPT-COF-Q and TFPA-TPB-COF-Q still kept the similar nanosphere morphology formed by stacking of multiple layers (Supplementary Figs. 1–6). TEM measurements were further carried out to the ultrasonicated samples. As shown in Fig. 2g, the TEM image of TFPA-TAPT-COF-Q clearly showed the ordered crystalline lattice fringes with an interplanar distance of $0.36 ± 0.01$ nm, which were closely associated with the vertical interlayer spacing of the simulated models. Moreover, the TEM image of TFPA-TPB-COF-Q clearly exhibited the honeycomb-shaped crystalline lattice (Fig. 2h), agreeing well with the simulated quinoline-linked 2D framework. The TEM images of both samples further confirmed the validity of the simulated crystal structures.

Fourier transform infrared (FT-IR) spectroscopy and solid-state CP-MAS [13]C NMR spectroscopy were then performed to test the bond formation in both samples of TFPA-TAPT-COF-Q and TFPA-TPB-COF-Q (Fig. 3a–f and Supplementary Figs. 7–10). In the FT-IR spectra of TFPA-TAPT-COF-Q (Fig. 3a and Supplementary Fig. 7), the pyridyl stretching frequency corresponding to the quinoline-rings at -1610 cm$^{-1}$ was observed, when a remarkable reduction of the feature peak at -1627 cm$^{-1}$ of the C = N group from Schiff base linkages in pristine TFPA-TAPT-COF was also detected, exhibiting the successful

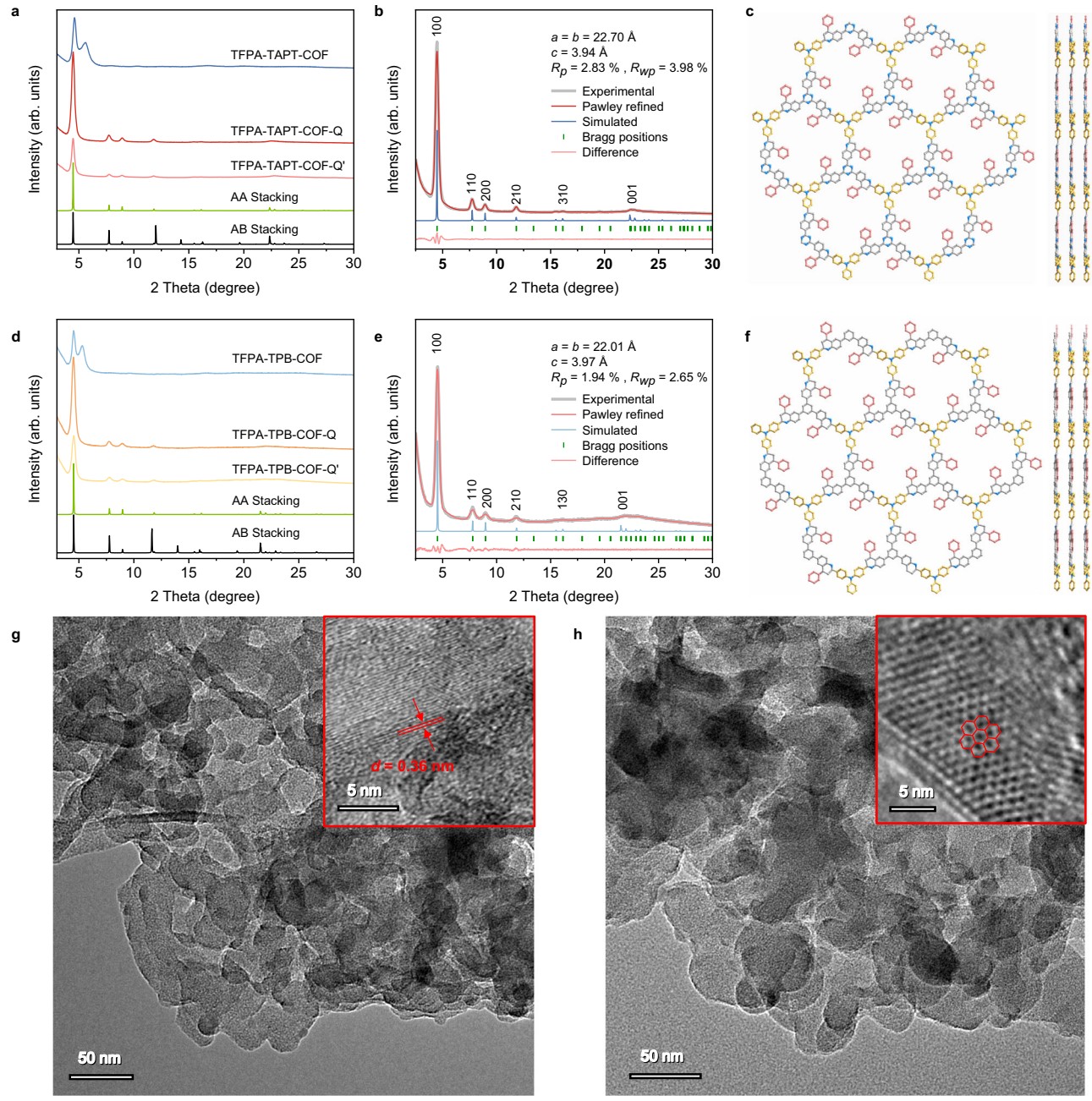

**Fig. 2 | PXRD patterns, simulated models and TEM images. a** PXRD patterns of TFPA-TAPT-COF (blue), TFPA-TAPT-COF-Q (red), and TFPA-TAPT-COF-Q' (light red), and simulated AA (green) and AB (black) stacking models for TFPA-TAPT-COF-Q. **b** PXRD patterns of TFPA-TAPT-COF-Q: experimental (grew), Pawley refined (red), simulated from its AA packing model (blue), and the differences between the Pawley refined and experimental patterns. **c** Perspective view (left) and side view (right) of TFPA-TAPT-COF-Q in AA stacking ball-and-stick model (H atoms are omitted for clarity). **d** PXRD patterns for TFPA-TPB-COF (light blue), TFPA-TPB-COF-Q (orange), and TFPA-TPB-COF-Q' (light orange), and simulated AA (green) and AB (black) stacking models for TFPA-TPB-COF-Q. **e** PXRD patterns of TFPA-TPB-COF-Q: experimental (grew), Pawley refined (red), simulated from its AA packing model (blue), and the difference between the Pawley refined and experimental patterns. **f** Perspective view (left) and side view (right) of TFPA-TPB-COF-Q in AA stacking ball-and-stick model (H atoms are omitted for clarity). TEM images of **g** TFPA-TAPT-COF-Q and **h** TFPA-TPB-COF-Q.

conversion of imine linkages into quinoline-ring during the modification reactions[38–51]. In the solid-state $^{13}$C NMR spectra (Fig. 3b and Supplementary Fig. 9), the feature carbon peak of -C = N- bonds at 158 ppm[53,54,59,60] in TFPA-TAPT-COF shifted to 156 ppm, which can be assigned to the signal of carbon in quinoline groups, further confirming the successful formation of the quinoline-ring in TFPA-TAPT-COF-Q[38–51]. X-ray photoelectron spectroscopy (XPS) measurements were also carried out to verify the chemical states of linked N atoms in these COFs. The peak of the nitrogen in the -C = N- imine fragment and triazine moieties at ~398.7 eV in TFPA-TAPT-COF shifted to lower

binding energy at ~398.6 eV with a broadened full width at half maxima (FWHM) after modification (Fig. 3c), indicating the success conversion of imine linkages to quinoline linkages. Moreover, XPS spectra showed that the signals of N 1 s in the triphenylamine fragments at 400.31 eV in the original TFPA monomer (Supplementary Figs. 11 and 12) slightly shifted to ~400.1 eV in the imine-linked COFs and ~399.9 eV after forming quinoline-linked COFs[38–51], which indicate that after the synthesis from TFPA monomer to COFs, the electron distribution of N atoms is broadened due to the formation of the donor-acceptor system in TFPA-TAPT-COF and donor-π-acceptor system in the quinoline-

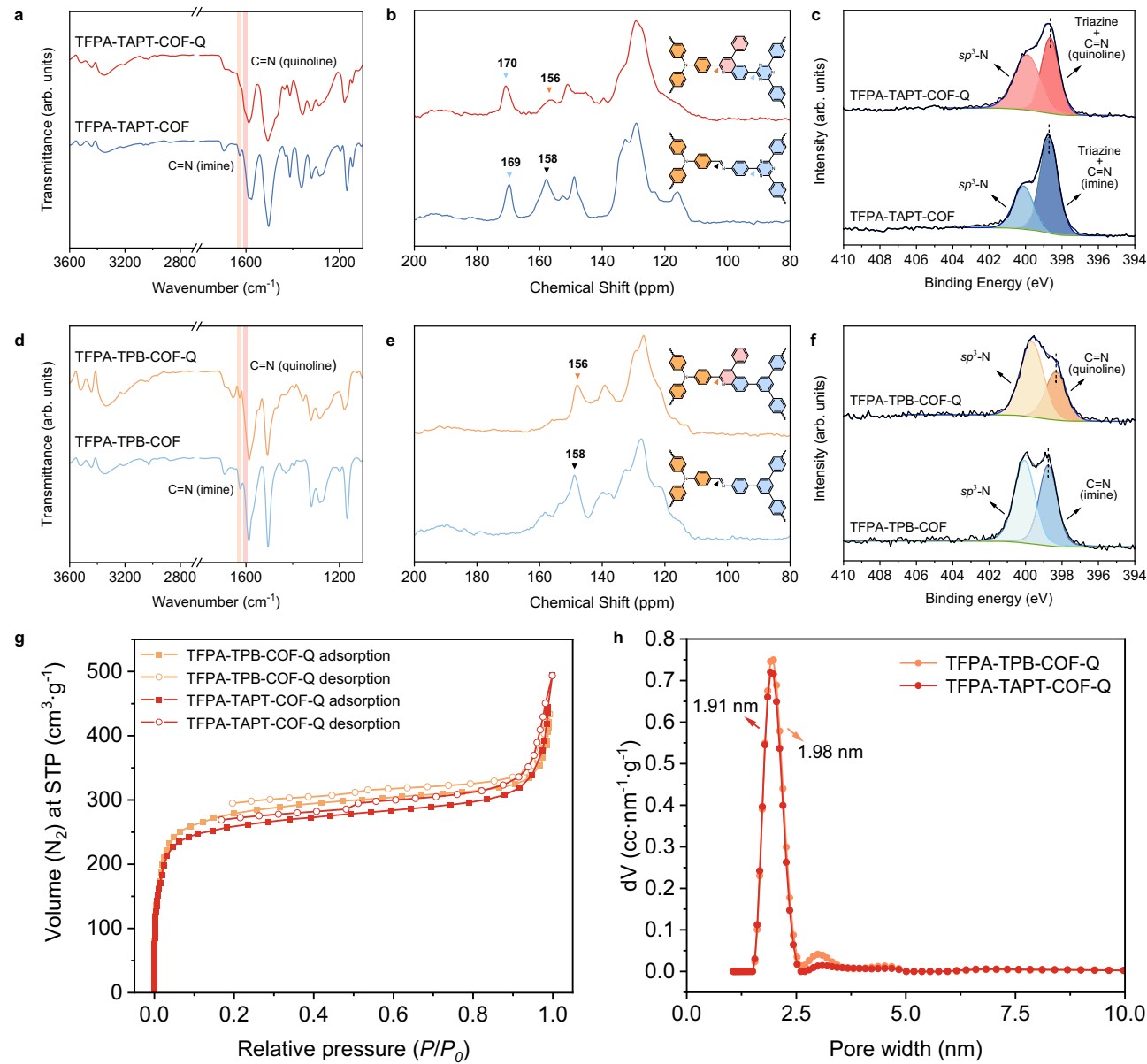

**Fig. 3 | Structural characterization data. a** FT-IR spectra and **b** $^{13}$C NMR spectra of TFPA-TAPT-COF-Q (red) and TFPA-TAPT-COF (blue). **c** XPS spectra of N 1 *s* in TFPA-TAPT-COF-Q and TFPA-TAPT-COF. **d** FT-IR spectra and **e** $^{13}$C NMR spectra of TFPA-TPB-COF-Q (orange) and TFPA-TPB-COF (light blue). **f** XPS spectra of N 1 *s* in TFPA-TPB-COF-Q and TFPA-TPB-COF. **g** N$_2$ sorption isotherms of TFPA-TAPT-COF-Q (red) and TFPA-TPB-COF-Q (orange) at 77 K. **h** Calculated pore size distributions of TFPA-TAPT-COF-Q (red) and TFPA-TPB-COF-Q (orange).

linked TFPA-TAPT-COF-Q lowering the binding energy of N 1 *s* in the triphenylamine fragments. These results also indicated that the imine bonds were successfully converted to substituted quinoline bonds upon modification. Similar phenomena were also detected in the FT-IR spectra, solid-state $^{13}$C NMR spectra, and XPS spectra of TFPA-TPB-COF-Q and TFPA-TAPT-COF (Fig. 3d–f as well as Supplementary Figs. 8 and 10), indicating the universality of the method in converting imine linkages into quinoline linkages in COFs.

After treating the quinoline-linked COFs of TFPA-TAPT-COF-Q and TFPA-TPB-COF-Q by Soxhlet extraction for 72 h, both samples were further activated by the degassing process at 120 °C for 12 h. The permanent porosity of the COFs was then investigated by N$_2$ sorption isotherm measurements at 77 K. The synthesized COFs exhibited a sharply increased step prior to the platform under relatively lower pressure ($P/P_0 < 0.04$) as evidenced by reversible type I sorption isotherm[61], demonstrating that they possess microporous features. The surface areas of TFPA-TAPT-COF-Q and TFPA-TPB-COF-Q

calculated by the Brunauer–Emmett–Teller (BET) method are 1058.99 and 1116.79 m$^2$ g$^{-1}$, with the pore size distribution values of ~1.91 and ~1.98 nm respectively (Fig. 3g, h), calculated based on the quenched solids density functional theory (QSDFT) method. The pore size distribution values agree well with the values calculated from their simulated crystal structures (~1.98 and ~1.97 nm for TFPA-TAPT-COF-Q and TFPA-TPB-COF-Q, respectively), further confirming the validity of the simulated crystal structures of the quinoline-linked COFs. Here, we should point out that, comparing with the related pristine imine-linked COFs, the BET surface areas of both quinoline-linked COFs decreased, but their pore size distribution values slightly increased (Supplementary Figs. 13–16). The smaller pore size distribution values, ~1.72 and ~1.78 nm for TFPA-TAPT-COF and TFPA-TPB-COF, respectively, should be caused by a stepwise slide of the 2D frameworks in their vertical direction[56–58], which also conversely indicate that the quinoline linkages in both modified COFs could enhance their longitudinal π-π stacking interactions and the crystallinity of quinoline-linked COFs.

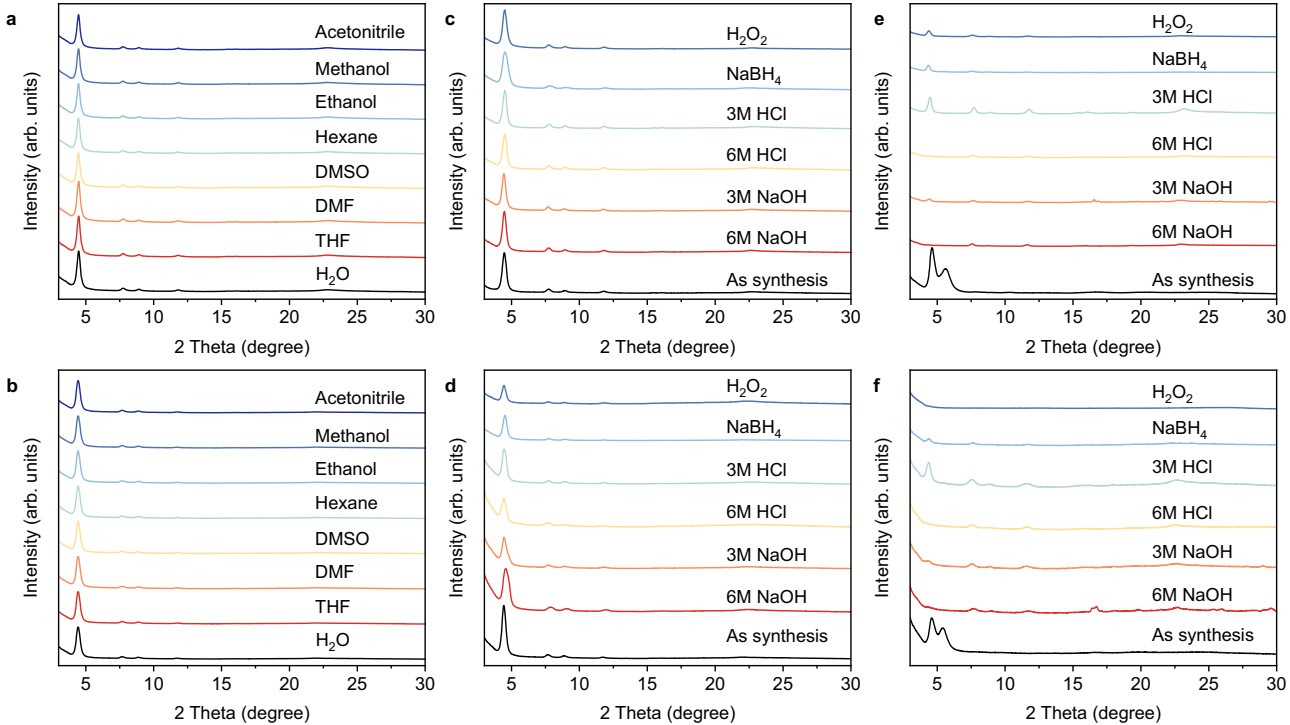

**Fig. 4 | PXRD patterns.** PXRD patterns of **a** TFPA-TAPT-COF-Q and **b** TFPA-TPB-COF-Q after socking in different organic solvents or water for 12 h. PXRD patterns of **c** TFPA-TAPT-COF-Q, **d** TFPA-TPB-COF-Q, **e** TFPA-TAPT-COF and **f** TFPA-TPB-COF after socking in different concentrations of acids, bases, $H_2O_2$ in ethanol and $NaBH_4$ in methanol for 12 h.

After successfully converting the imine linkages into quinoline linkages in both COFs, their thermal and chemical stabilities were then studied. Thermogravimetric analysis (TGA) results measured under $N_2$ atmosphere showed that, as their pristine imine-linked COFs of TFPA-TAPT-COF and TFPA-TPB-COF, TFPA-TAPT-COF-Q and TFPA-TPB-COF-Q display high thermal stability up to ~400 °C, which is thermally stable enough for common photocatalytic aerobic oxidation reactions (Supplementary Figs. 17–20). PXRD measurements after socking into different solvents were then applied to characterize the chemical stability of the synthesized COFs. After immersing the samples in different common organic solvents, such as acetonitrile, methanol, ethanol, hexane, DMSO, DMF, and tetrahydrofuran as well as water for 12 h, the powder samples of both quinoline-linked COFs were recollected and dried in a vacuum oven and then their PXRD spectra were determined. The main peaks in the PXRD patterns of all these immersed samples were still well maintained (Fig. 4a, b), suggesting that the synthesized quinoline-linked COFs also embody high chemical stability in common organic solvents as their pristine imine-linked COFs of TFPA-TAPT-COF and TFPA-TPB-COF (Supplementary Figs. 21 and 22).

Subsequently, the chemical stability of the quinoline-linked samples after immersion in aqueous solutions with different concentrations of acids and bases and alcohol solutions containing oxidizing and reducing agents was studied by the PXRD measurements. No matter immersed in 3 and 6 mol/L of HCl or NaOH solutions, ethanol solution containing $H_2O_2$, or methanol solution containing $NaBH_4$ for 12 h, the main peaks of the PXRD patterns for TFPA-TAPT-COF-Q and TFPA-TPB-COF-Q were still maintained well (Fig. 4c, d). For TFPA-TAPT-COF and TFPA-TPB-COF, although the protonation phenomena as previously reported[56–58] was observed in relatively low concentration of acid (Supplementary Figs. 23–25), a remarkable decrease of the main peak in their PXRD patterns was detected after long-term soaking in a higher concentration of acid, and the main peaks in their PXRD patterns sharply decreased after immersing in either low or high concentration of base solutions. The same results were also observed in their PXRD patterns of imine-linked COFs after immersing in alcohol solutions containing oxidizing and reducing agents (Fig. 4e, f). The stability investigations revealed that after converting the imine linkages into quinoline linkages, TFPA-TAPT-COF-Q and TFPA-TPB-COF-Q can maintain their chemical stability well in harsh acid, base and oxidizing, and reducing agent solutions.

Due to the photoactive TPA groups being incorporated in both quinoline-linked COFs, the photochemical properties of TFPA-TAPT-COF-Q and TFPA-TPB-COF-Q were further investigated (Fig. 5a–f). Solid-state UV−vis diffuse reflectance spectrum (DRS) measurements were performed and wide absorption bands from ~250 to 600 nm in both quinoline-linked COFs were observed, which exhibited broader absorption ranges than corresponding pristine imine-linked COFs (TFPA-TAPT-COF and TFPA-TPB-COF), agreeing well with their related color change from orange to brown after modifying by quinoline groups (Fig. 5a, d). After Kubelka−Munk transforming based on the results of solid-state UV−vis DRS measurements[62–64], the corresponding optical bandgaps of TFPA-TAPT-COF-Q and TFPA-TPB-COF-Q were calculated to be 1.73 and 1.80 eV respectively, while the bandgap values of their related pristine imine-linked COFs are 2.25 and 2.42 eV, indicating that the quinoline-modification can reduce the width of their bandgap and increase the utilization range of light effectively (Fig. 5b, e). After obtaining their bandgap values, further calculations about the flat-band potential (FBP) were then carried out. Based on their Mott−Schottky tests in 0.5 M $Na_2SO_4$, their FBP values were measured under different frequencies with the three-electrode system, in which Ag/AgCl electrode was used as the reference electrode. The results are −0.96 and −0.92 V (Fig. 5c, f) for TFPA-TAPT-COF-Q and TFPA-TPB-COF-Q respectively, indicating that their conduction bands based on normal hydrogen electrode (NHE) were −0.76 and −0.72 V, respectively. Furthermore, their valence band values were calculated by combining with the optical bandgap values obtained from the Kubelka−Munk transformed reflectance spectra and Mott−Schottky tests (Fig. 5g). According to the equation

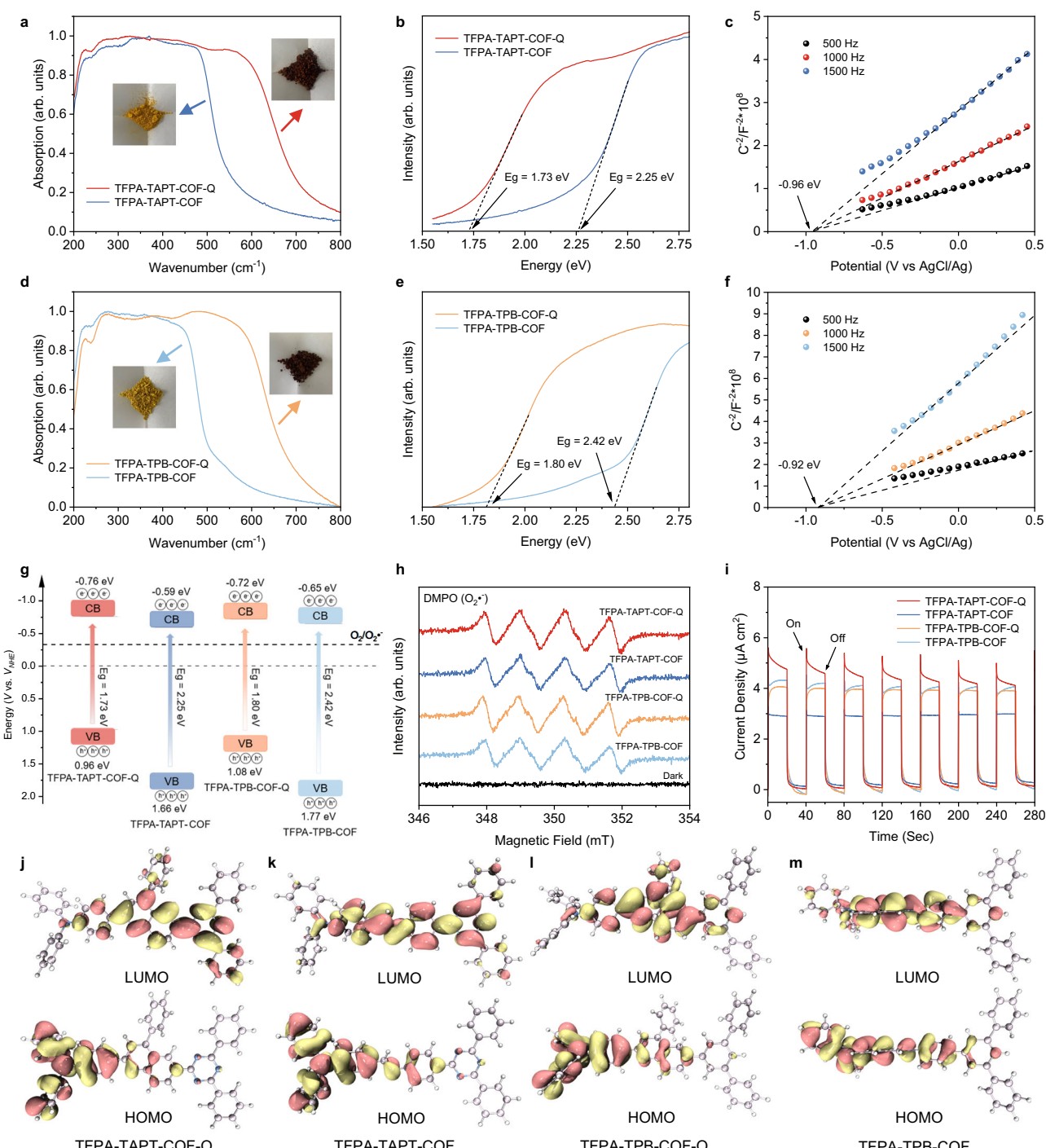

**Fig. 5 | Photochemical property studies. a** UV-Vis DRS spectra and **b** Kubelka-Munk-transformed reflectance spectra based on the UV-Vis spectra of TFPA-TAPT-COF-Q (red) and TFPA-TAPT-COF (black). **c** Mott-Schottky curve of TFPA-TAPT-COF-Q. **d** UV-Vis DRS spectra and **e** Kubelka-Munk-transformed reflectance spectra based on the UV-Vis spectra of TFPA-TPB-COF-Q (red) and TFPA-TPB-COF (black). **f** Mott-Schottky curve of TFPA-TPB-COF-Q. **g** Calculated band alignments, **h** EPR

spectra in MeCN under dark (black) and the irradiation of the Xe lamp with DMPO as the trapping agent and **i** transient photocurrent response curves of TFPA-TAPT-COF-Q (red), TFPA-TAPT-COF (blue), TFPA-TPB-COF-Q (yellow) and TFPA-TPB-COF (light blue). Calculation of HOMO/LUMO for the structural segments of **j** TFPA-TAPT-COF-Q, **k** TFPA-TAPT-COF, **l** TFPA-TPB-COF-Q and **m** TFPA-TPB-COF.

$E_{CB} = E_{VB} - E_g$, the valence band values ($V$ vs. $V_{NHE}$) are 0.96 and 0.88 V for TFPA-TAPT-COF-Q and TFPA-TPB-COF-Q respectively, while the pristine imine-linked COFs are 1.66 eV and 1.77 eV (Fig. 5a, b, d, e, and Supplementary Figs. 26 and 27). Thus, quinoline-linked COFs exhibited typical photochemical properties of semiconductors, indicating that the synthesized quinoline-linked COFs are highly promising candidates for photocatalytic aerobic oxidations.

TPA-containing materials usually can initiate the generation of reactive oxygen species in the presence of oxygen or air. Therefore, the ability to generate the ROS by the synthesized quinoline-linked COFs under light irradiation was then detected by electron paramagnetic resonance (EPR) measurements[65,66]. Compared to the electric potential of $O_2/O_2^{•-}$ ($-0.33$ $V$ vs. $NHE$)[67], both the energy gap of the two kinds of COFs can cover it, revealing that they have enough driving

force for the photocatalytic reduction of $O_2$ to $O_2^{•−}$ (Fig. 5h). Therefore, 5,5-dimethyl-1-pyrroline N-oxide (DMPO) as the commonly used spin trapping agent for $O_2^{•−}$ was added into the suspensions of the quinoline-linked COFs in MeCN and then they were illuminated by a Xe lamp for 10 min in the atmospheric environment. Based on EPR measurements, the typical multiple peaks of $O_2^{•−}$ were observed clearly in the EPR curves of the samples with the quinoline-linked COFs as photocatalysts when compared with the results obtained with the pristine imine-linked COFs as photocatalysts. Contrarily, no signal was detected by EPR measurements under a dark environment (Fig. 5h) with both samples as photocatalysts. Meanwhile, the same measurements were also applied to detect the production of singlet oxygen ($^1O_2$) with 2,2,6,6-tetramethylpiperidine (TEMP) as the commonly used spin trapping agent. The EPR measurement results revealed that no signal of $^1O_2$ was detected no matter under light illumination or not (Supplementary Figs. 28 and 29). These results clearly revealed that the quinoline-linked COFs can trigger the generation of $O_2^{•−}$ under light illumination in the presence of the oxygen molecule.

Transient photocurrent response measurements were also used to investigate their effectiveness toward photocurrent response, and the electron-hole separation and charge-carrier transfer properties were observed under light irradiation for the quinoline-linked COFs. Continuous and intense photocurrent responses were recorded in both samples from the light switching on/off during a certain interval. Moreover, the photocurrent intensity of TFPA-TAPT-COF-Q was remarkably enhanced as compared with the pristine imine-linked TFPA-TAPT-COF, which indicated that the separation efficiency of photocatalytically generated electrons and holes by COFs increased and the conversion of imine linkages into quinoline linkages can enhance their capability to capture photons to produce excited state carriers (Fig. 5i). For TFPA-TPB-COF-Q, its photocurrent intensity even decreased a little bit when compared with the pristine imine-linked TFPA-TPB-COF.

In both quinoline-linked COFs, the only differences between them are the TAPT and TPB organic modules. Although the TAPT and TPB modules are quite similar in size and symmetry, the constructed COFs, TFPA-TAPT-COF-Q and TFPA-TPB-COF-Q, displayed a large difference in their photocurrent intensity. In order to explain this difference, density functional theory (DFT) calculations on their electron structures were carried out. We speculated that TFPA and TAPT play the roles of electron-donor and electron-acceptor, respectively, to form a D-A structure that could not be formed by TPB monomer. After converting the imine linkages into quinoline linkages, the π-conjugate system is expended to give the formation of a D-π-A structure[38–45], which increases the separation efficiency of photocatalytically generated electrons and holes. To demonstrate this deduction, the core fragments of both quinoline-linked COFs with their related imine-linked COFs were then constructed to determine the orbital distribution of electrons in the highest occupied molecular orbital (HOMO) and the lowest unoccupied molecular orbital (LUMO) by DFT calculations (Fig. 5j–m) with the basis set of b3lyp/6-311 g + (d, p)[68,69]. According to the calculated results, TFPA-TAPT-COF-Q showed a more dispersed electron distribution in LUMO as compared to TFPA-TAPT-COF (Fig. 5j, k), which should be ascribed to the situation that the formed quinoline linkages increased the π-conjugated structure and finally promoted the charge-carrier transfer process. Nevertheless, the TPB moiety in the TFPA-TPB-COF-Q cannot easily display the electron-acceptor role, exhibiting a narrow distribution to LUMO as compared to the triazine center in TAPT moiety (Fig. 5l, m)[70]. The DFT calculations well explained the higher photocurrent intensity of TFPA-TAPT-COF-Q than TFPA-TPB-COF-Q, revealing the TAPT-incorporated TFPA-TAPT-COF-Q is a more active photocatalyst for photocatalytic aerobic oxidations.

While the quinoline-linked TFPA-TAPT-COF-Q displays a higher photocurrent intensity than TFPA-TPB-COF-Q, they show positive results in photochemical properties, demonstrating that the synthesized quinoline-linked COFs should be effective photocatalysts for aerobic oxidations. As the quinoline-linked COFs possess higher photo-responsive properties, their photocatalytic aerobic oxidations were then investigated. Photocatalytic oxidation of sulfide as a typical reaction with the process by photocatalytically generating $O_2^{•−}$ is often taken as a model reaction to detect the activity of the photocatalysts[19–22,71,72]. By taking TFPA-TAPT-COF-Q as the catalyst, we first investigated its photocatalytic activity on aerobic oxidation of sulfide with methylphenyl sulfide as the representative substrate (Supplementary Table 2). Then, we also measured other synthesized COFs under the same optimal conditions. The studies revealed that all of the four synthetic COFs displayed good recyclable photocatalytic performance (Supplementary Figs. 30–43). In addition, their high photocatalytic performance still possessed when extending the substrates to other sulfides (Supplementary Tables 3 and 4) upon photocatalytically generating $O_2^{•−}$ (Supplementary Fig. 44).

The high photocatalytic activity of TFPA-TAPT-COF-Q combing with their high stability encouraged us to test their efficiency and recyclability for photocatalytic aerobic oxidations in harsh conditions. The decarboxylation reaction of carboxylic acids was one of the most meaningful transformations in organic synthesis due to the products being widely applied as structural moieties and starting materials in organic materials synthesis[73–76]. However, oxidative decarboxylation of arylacetic acids was usually carried out in the presence of alkaline cocatalysts and catalyzed by metal complexes such as Mn (III) or Cu (II) based compounds in homogeneous reaction systems, which usually encounter with stability issues of catalysts as well as environmental pollution problems[73–76]. As a highly stable metal-free heterogeneous photocatalyst, the catalytic activity and reusability of TFPA-TAPT-COF-Q for decarboxylation reaction of carboxylic acids were studied. Taking 4-methoxyphenylacetic acid as the representation reactant, the decarboxylation reaction was conducted under different conditions (Entries 1–11, Supplementary Table 5). According to the literature report[73], 1,1,3,3-tetramethylguanidine (TMG) was used as the cocatalyst in this reaction. In the presence of the TMG cocatalyst, the 4-methoxyphenylacetic acid reached a complete decarboxylation within 9 h. In order to compare the effect of different bases, a variety of different organic and inorganic bases were also attempted (Entries 12–15, Supplementary Table 5) and the experimental results confirmed that TMG is the best choice (Entry 6, Supplementary Table 5). Moreover, 4-methoxybenzaldehyde was detected as the main product, although two products of 4-methoxybenzaldehyde and 4-methoxybenzyl alcohol can be both formed theoretically.

In practical applications, reusability is a significant factor of heterogeneous catalysis, especially in harsh conditions. Taking TFPA-TAPT-COF-Q as a photocatalyst, recycle photocatalytic reactions for oxidative decarboxylation of arylacetic acids were carried out, and the results clearly revealed that no obvious decrease was detected in conversion rates after five cycles of reactions (Fig. 6a). In contrast, for TFPA-TAPT-COF, the suspension became nearly clear after only the first run, which cannot be recycled for further use (Supplementary Fig. 45). PXRD measurement was applied on the recycled sample of TFPA-TAPT-COF-Q, showing that the structure of TFPA-TAPT-COF-Q was still maintained after the five runs of catalytic reactions (Fig. 6b). FT-IR spectra were recorded to examine the framework stability of TFPA-TAPT-COF-Q, and the feature peaks in the spectra of the samples after the reactions agreed well with those of the as-synthesized TFPA-TAPT-COF-Q, proving the stability of the framework (Fig. 6c). XPS tests were also conducted for TFPA-TAPT-COF-Q before and after the catalytic experiments. The fine spectra of N 1$s$ for TFPA-TAPT-COF-Q before and after oxidative decarboxylation of arylacetic acids were consistent well with each other, indicating that the formed quinoline linkage in TFPA-TAPT-COF-Q was well maintained after the five runs of catalytic reactions (Fig. 6d). Similar measurements were also carried

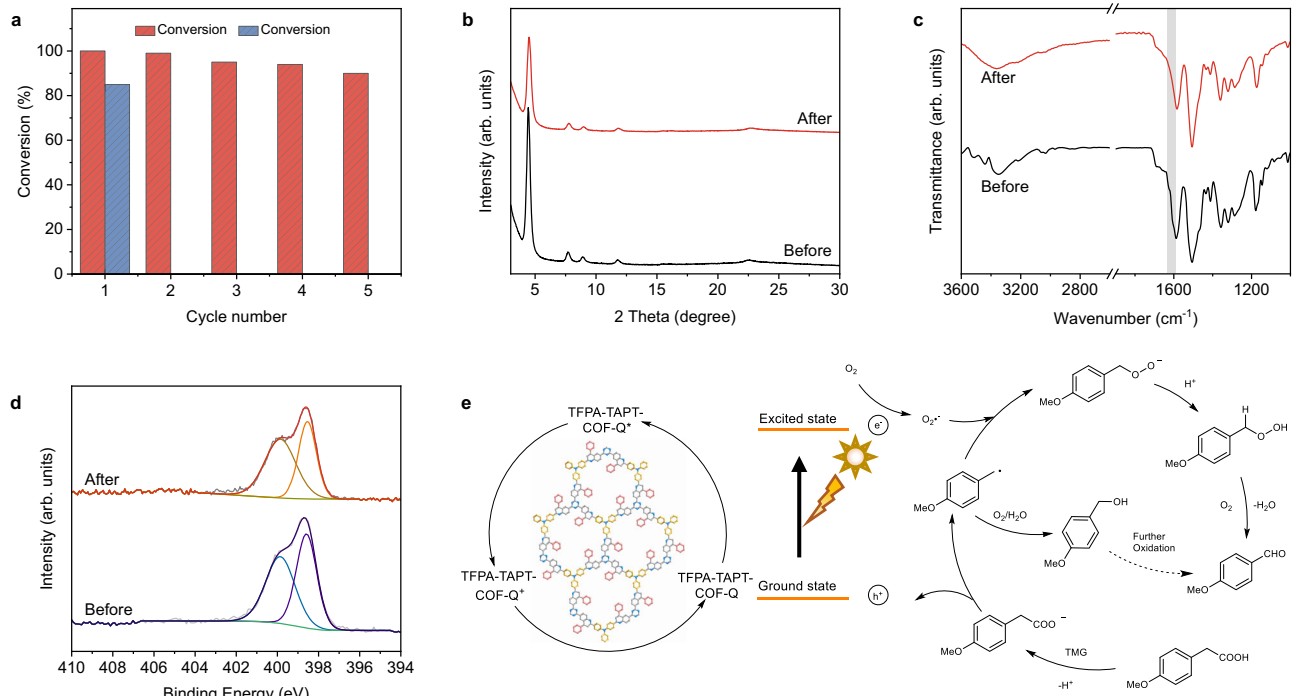

**Fig. 6 | Oxidative decarboxylation of arylacetic acids. a** Five runs of oxidative decarboxylation of arylacetic acids with TFPA-TAPT-COF-Q (red) and TFPA-TAPT-COF (blue) as photocatalysts. **b** PXRD pattern, **c** FT-IR spectra, and **d** N 1*s* XPS spectra for TFPA-TAPT-COF-Q before and after photocatalytic reactions. **e** Proposed mechanism for the decarboxylation of arylacetic acids.

out for TFPA-TPB-COF-Q, and the same outcomes as the first cycle of reaction were detected in all five recycling runs although its photo-activity is not as high as that of TFPA-TAPT-COF-Q (Supplementary Figs. 46–49). These results demonstrated that the quinoline-linked COFs, TFPA-TAPT-COF-Q and TFPA-TPB-COF-Q, can be used as effective recyclable heterogeneous photocatalysts in relatively harsh organic acid conditions.

Then, we extended the substrates to a wide scope under the stirring in MeCN for 9 h by adding TMG as a standard condition. As shown in Table 1, when the para-position of arylacetic acids was substituted by either electron-donating groups such as -Me and −C(CH)₃ (Entries 2 and 3, Table 1) or electron-withdrawing groups such as -F, -Cl, and -Br (Entries 4–6, Table 1) and even changed to other substituents such as naphthyl group (Entry 7, Table 1), the quinoline-linked TFPA-TAPT-COF-Q also displayed higher photocatalytic activity in conversion rates as compared with the imine-lined TFPA-TAPT-COF. The main product was also corresponding benzaldehyde. The same results were observed when replacing the photocatalyst by TFPA-TPB-COF-Q (Supplementary Figs. 50 and 51). Thus, the results confirmed that the quinoline-linked COFs not only can keep stable under relatively harsh conditions, but also display higher photoactivity in photocatalytic oxidative decarboxylation of arylacetic acids.

As the quinoline-linked TFPA-TAPT-COF-Q has a high photocatalytic efficiency in the decarboxylation of arylacetic acids, the reaction mechanism was then explored (Entries 19–23, Supplementary Table 5). At the aforementioned optimal reaction condition (Entries 19–21, Supplementary Table 5), three different trapping agents, benzoquinone (BQ) for $O_2^{\bullet-}$, diazabicyclo[2.2.2]octane (DABCO) for $^1O_2$ and potassium iodide (KI) for electrons, were added to verify the mechanism of the reaction. The conversion rate in the presence of KI additive showed that the TFPA-TAPT-COF-Q⁺ holes play a key role during the reaction according to entry 19 in Supplementary Table 5. Puzzlingly, after the addition of the consuming reagent of $O_2^{\bullet-}$ (BQ), the conversion rate still retained up to 62% (Entry 20, Supplementary Table 5), while the main product transformed from

4-methoxybenzaldehyde to 4-methoxybenzyl alcohol under aforementioned optimal condition. Thus, further investigations were applied by the addition of the external proton source and changing the gas atmosphere, respectively. After adding water in MeCN as the co-solvent, the aldehyde selectivity was sharply decrease to 12%, exhibiting that there were almost no aldehyde products in the presence of water (Entry 22, Supplementary Table 5). When we changed the air to pure oxygen as the gas atmosphere in the absence of water, the product becomes only aldehyde (Entry 23, Supplementary Table 5), demonstrating that the water molecule from the air or solvents in the reaction system is the main reason to lead to the generation of alcohol product. These studies revealed that photocatalytic process is conducted as that after the photoexcitation of the TFPA-TAPT-COF-Q to form TFPA-TAPT-COF-Q*, the photogenerated electrons separated and generated TFPA-TAPT-COF-Q⁺ holes. Then the TMG activated substrates are first oxidized by the generated TFPA-TAPT-COF-Q⁺ holes to give the formation of a phenylmethyl radical, which is detected in the oxidative decarboxylation of arylacetic acids (Supplementary Fig. 52). Subsequently, the products are produced in two different ways based on the presence of external proton source or not. A proposed reaction mechanism is shown in Fig. 6e[73–76].

The coupling reaction of benzylamine is also a typical aerobic oxidation[77]. The inevitable existence of the amine group in the substrates can probably destroy the reversible imine group in Schiff base linked COFs, making the reaction relatively harsh. Thus, the coupling reaction of benzylamine was also selected as a verification reaction to detect the stability of fabricated quinoline-linked COFs. Taking TFPA-TAPT-COF-Q as a catalyst, after the screening of multiple experimental conditions (Entries 1–10, Supplementary Table 6), we discovered that the benzylamine can be fully converted to the product within 3.5 h in MeCN (Entry 6, Supplementary Table 6). Then, the yield of the product did not change anymore even prolonging the reaction time (Entry 7, Supplementary Table 6). Interestingly, when replacing the photo-catalyst by TFPA-TAPT-COF under the same reaction conditions, a complete conversion was also achieved (Entry 11, Supplementary

**Table 1 | Photocatalytic results of TFPA-TAPT-COF-Q and TFPA-TAPT-COF in oxidative decarboxylation of arylacetic acids (TMG = 1,1,3,3-tetramethylguanidine)**

| Entry | Substrate | TFPA-TAPT-COF-Q | | TFPA-TAPT-COF | |
|---|---|---|---|---|---|
| | | Conv. (%) | Sel. (%) | Conv. (%) | Sel. (%) |
| 1 | MeO-substituted | 100 | 86 | 85 | 76 |
| 2 | methyl-substituted | 78 | 81 | 72 | 78 |
| 3 | t-Bu-substituted | 77 | 80 | 44 | 77 |
| 4 | F-substituted | 64 | 84 | 56 | 75 |
| 5 | Cl-substituted | 83 | 92 | 36 | 77 |
| 6 | Br-substituted | 85 | 92 | 70 | 82 |
| 7 | naphthyl | 100 | 84 | 100 | 84 |

Table 6). Lower conversion rates were detected when changing the catalyst to TFPA-TPB-COF-Q and TFPA-TPB-COF under the same conditions (Entries 12 and 13, Supplementary Table 6).

About the reusability in the coupling reaction of benzylamine, similar results as observed above were achieved (Supplementary Figs. 53–57), that is, TFPA-TAPT-COF-Q could still keep its high photocatalytic activity within the five cycles of reactions (Fig. 7a), while a great loss in mass for the recovered powder of TFPA-TAPT-COF after each cycle was detected, which cannot be recollected after the third cycle of reaction (Supplementary Fig. 53). The PXRD measurements on the recollected sample of TFPA-TAPT-COF-Q revealed that the main peaks were still maintained after five runs of reactions (Fig. 7b).

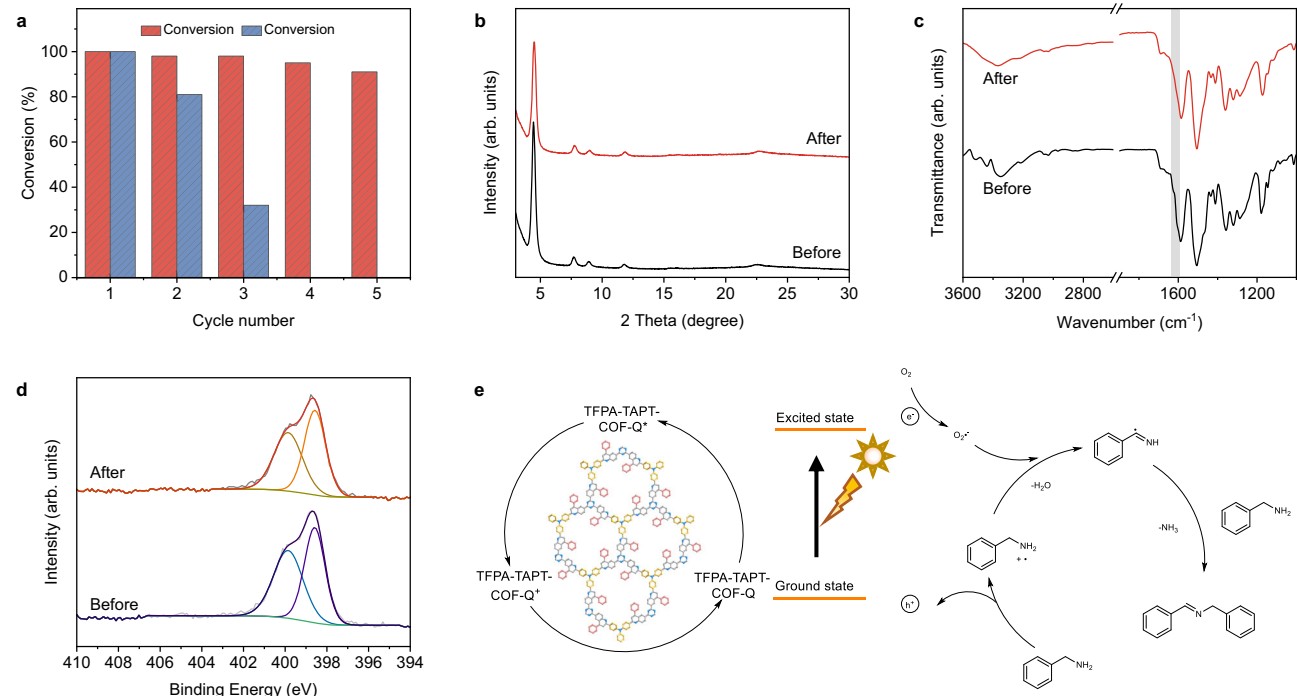

**Fig. 7 | Benzylamine coupling reactions. a** Five runs of benzylamine coupling with TFPA-TAPT-COF-Q (red) and TFPA-TAPT-COF (blue) as the photocatalyst. **b** PXRD patterns, **c** FT-IR spectra and **d** N 1*s* XPS spectra for TFPA-TAPT-COF-Q before and after photocatalytic reactions. **e** Proposed coupling mechanism of benzylamine.

Meanwhile, FT-IR and XPS spectra were also employed to check the framework stability of TFPA-TAPT-COF-Q. The spectra of recollected samples of TFPA-TAPT-COF-Q (Fig. 7c, d) showed that they were consistent well with the as-synthesized sample before carrying out the coupling reaction of benzylamine. The results not only indicated the high stability of the quinoline-linked TFPA-TAPT-COF-Q, but also confirmed that the photoactive COF can be used as an effective recyclable photocatalyst in relatively harsh base conditions.

Afterward, the photocatalytic activity of TFPA-TAPT-COF-Q was detected by enriching the scope of the substrates. When reacted under the same optimal conditions as the coupling reaction of benzylamine, for different substituted benzylamines, no matter para-position substituted by electron-donating groups such as -Me, -C(CH)$_3$ and -OMe (Entries 2–4, Table 2) or electron-withdrawing groups such as -F, -Cl and -Br (Entries 5–7, Table 2), all of them can give higher yields comparing with the imine-linked TFPA-TAPT-COF as the catalyst (Table 2 and Supplementary Figs. 58 and 59). The same results were also obtained when replacing the catalyst by TFPA-TPB-COF-Q and TFPA-TPB-COF (Supplementary Figs. 54–57), showing that the quinoline linkage in these photoactive COFs not only enhanced the robustness of the frameworks, but also increased the photoactivity of the formed frameworks. The possible mechanism of coupling reactions was investigated by adding the consuming reagents into the reaction system (Entries 14–16, Supplementary Table 6)[77]. According to the obtained results, the reaction mechanism for the photocatalytic coupling of benzylamine is shown in Fig. 7e.

The effective production of H$_2$O$_2$ from water under air atmosphere by heterogeneous photocatalysis is a green method in chemical industry due to the safety risks during the long-term storage and transportation of H$_2$O$_2$[48,51,78–84]. Because the generated H$_2$O$_2$ is a strong oxidant, the photocatalysts usually should be robust enough in such a strong oxidation condition. Inspired by the high stability in aforementioned strong oxidizing condition and high photocatalytic activity in both photocatalytic reactions of organic acid involving oxidative decarboxylation and organic base involving benzylamine coupling, we then investigated the photocatalytic properties of the

prepared quinoline-linked COFs under such a strong oxidation condition for photocatalytic production of H$_2$O$_2$. The quinoline-linked TFPA-TAPT-COF-Q was first chosen as a photocatalyst to explore the optimal reaction conditions. As shown in Fig. 8a, there was no H$_2$O$_2$ production detected under illumination in the absence of air, while an appropriate amount of H$_2$O$_2$ was generated in the presence of air, indicating that oxygen in air is the oxygen source for producing H$_2$O$_2$. When a verity of alcohols such as ethanol (EtOH), isopropanol (IPA), and triethanolamine (TEOA) were added into the aqueous solution as hole sacrificial reagents, obvious increases in the H$_2$O$_2$ production yields were observed. When changing the hole sacrificial reagents from the tested aliphatic alcohols to benzyl alcohol (BA), a further increase in H$_2$O$_2$ production yield up to 11831.6 µmol·g$^{-1}$·h$^{-1}$ within the first 1 h was observed (Fig. 8b). At the same condition, a value of 9251.0 µmol·g$^{-1}$·h$^{-1}$ for H$_2$O$_2$ production within 1 h was also obtained using TFPA-TPB-COF-Q as the photocatalyst and BA as hole sacrificial reagent (Fig. 8b), making these quinoline-linked COFs among the best COF-based photocatalysts for photocatalytic production of H$_2$O$_2$ (Supplementary Table 7)[48,51,78–84]. Comparing with other alcohols, higher efficiency of BA should be attributed to the strong π-π stacking interaction between the aromatic region of BA and benzene side chain in the substituted quinoline linkages of the COFs, thus facilitating the electrons transfer to the surface of quinoline-linkage COFs for achieving a higher hole sacrifice rate. For both of the imine-linked TFPA-TAPT-COF and TFPA-TPB-COF, remarkable decreases in the photocatalytic activity for the production of H$_2$O$_2$ under the same conditions were observed as shown in Fig. 8b. By considering the differences of their structures before and after the modification, the main reason should be that when converting imine linkages into quinoline linkages, the original donor-acceptor system in imine-linked COFs was changed into donor-π-acceptor system, which can increase the conjugation, extend the photo-absorption capability and finally improve the performance in photocatalysis. When compared with other reported COF-based photocatalysts possessing similar structures (Supplementary Table 7), higher

**Table 2 | Photocatalytic results of TFPA-TAPT-COF-Q and TFPA-TAPT-COF in coupling of benzylamines**

| Entry | Substrate | TFPA-TAPT-COF-Q Conv. (%) | TFPA-TAPT-COF Conv. (%) |
|---|---|---|---|
| 1 |  | 100 | 100 |
| 2 |  | 98 | 96 |
| 3 |  | 100 | 100 |
| 4 |  | 65 | 53 |
| 5 |  | 81 | 70 |
| 6 |  | 96 | 81 |
| 7 |  | 85 | 81 |

performance of both TFPA-TAPT-COF-Q and TFPA-TPB-COF-Q should also be attributed to the introduction of the photoactive TPA moieties into the COF skeletons.

Motivated by the high efficiency of the quinoline-linkage COFs in photocatalytic production of $H_2O_2$, their long-term photocatalytic capacity and recyclability were carried out to verify the robustness against the strong oxidizing environment (Fig. 8c, d). As shown in Fig. 8c, under continuous illumination, the $H_2O_2$ production yields

with TFPA-TAPT-COF-Q and TFPA-TPB-COF-Q as photocatalysts can increase steadily up to 295900 $\mu mol \cdot g^{-1}$ and 288090 $\mu mol \cdot g^{-1}$ over time respectively, while the pristine imine-linked COFs are easily decomposed to leave only a small amount of solids under the same conditions after about 24 h continuous reactions. Moreover, the photocatalytic activity of TFPA-TAPT-COF-Q and TFPA-TPB-COF-Q was not decreased even after 5 cycles of reactions based on the detection results of recollected samples for every 6 h per time (Fig. 8d). The

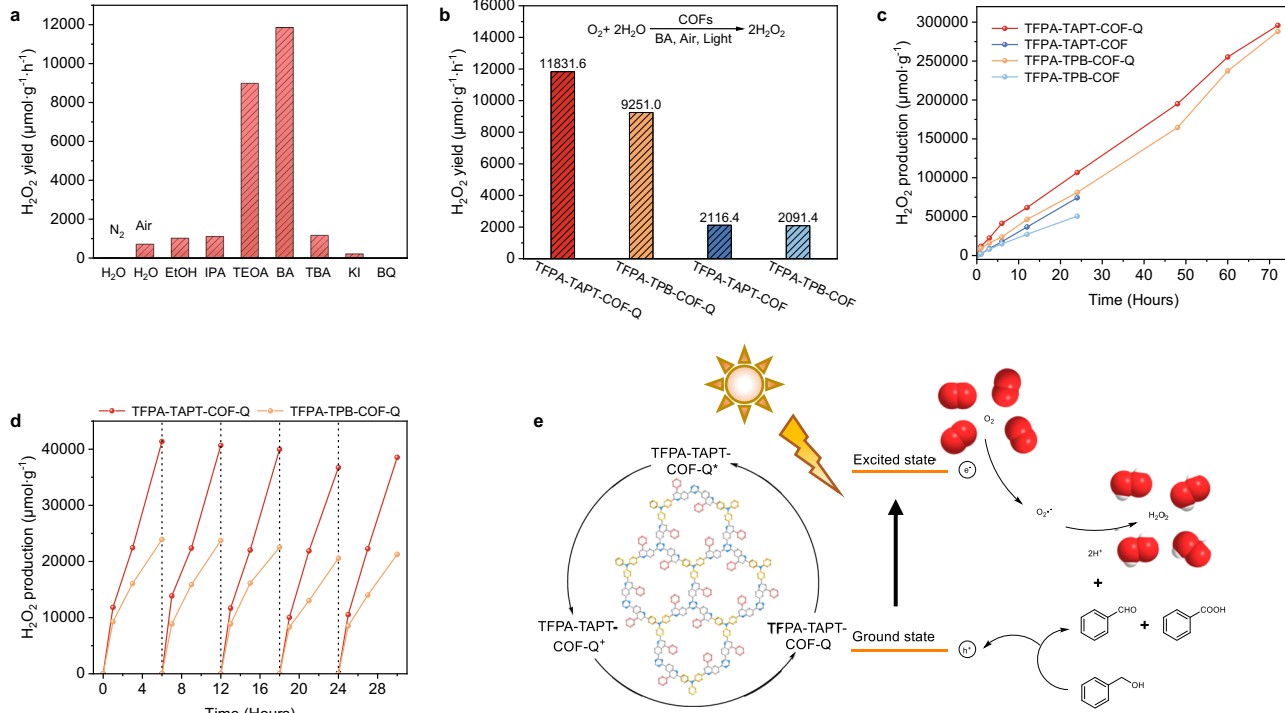

**Fig. 8 | Photocatalytic production of H₂O₂. a** Production yields of H₂O₂ with TFPA-TAPT-COF-Q as the photocatalyst in water and different sacrificial agents (10 mg of COF in 22 mL water or water/alcohol (10:1) solution at 25 °C for 1 h). **b** H₂O₂ production yields with photocatalysts of TFPA-TAPT-COF-Q (red), TFPA-TPB-COF-Q (orange), TFPA-TAPT-COF (blue) and TFPA-TPB-COF (light blue) in water/BA (10:1) solution. **c** Long-term photocatalytic H₂O₂ production of each COF in water/BA (10:1) solution. **d** Five runs of photocatalytic H₂O₂ production of TFPA-TAPT-COF-Q (red) and TFPA-TPB-COF-Q (orange) in water/BA (10:1) solution. **e** Proposed mechanism for photocatalytic production of H₂O₂.

PXRD measurements of the recollected samples after 24 h reactions also confirmed that the crystallinity of the quinoline-linked TFPA-TAPT-COF-Q and TFPA-TPB-COF-Q was still maintained, while main peaks in the PXRD patterns of the imine-linked COFs completely disappeared after 12 h reactions (Supplementary Figs. 60–63).

Finally, the mechanism for photocatalytic H₂O₂ production was investigated by quenching experiments (Fig. 8a). When tert-butyl alcohol (TBA) replacing the sacrificial agent of BA as the trapping agent for hydroxyl radical (•OH) was added to the mixture, the H₂O₂ production yield was found nearly the same as the case in the absence of the sacrificial agent. When KI or BQ as the typical trapping agent for O₂•⁻ was added to the mixture solvent of water and BA, the yields decreased sharply or even no H₂O₂ production was detected. These results suggest that the photocatalytic formation of O₂•⁻ by O₂ binding with electrons plays a key role in the reaction rather than •OH. Meanwhile, the benzaldehyde and benzoic acid were also detected in the liquid phase, indicating that BA was oxidized by photogenerated holes (Fig. 8e). Finally, the O₂•⁻ radical combines with the lost protons to give the formation of H₂O₂ (Supplementary Fig. 64)[48,51,78–84].

## Discussion

In summary, by converting the imine linkages into quinoline linkages in triphenylamine-incorporated photoactive COFs, two highly stable crystalline COFs, TFPA-TAPT-COF-Q and TFPA-TPB-COF-Q, have been successfully synthesized. The detailed characterizations revealed that quinoline linkages in the COFs not only enhanced the framework stability, but also increased their photochemical properties. Photocatalytic experiments confirm that the synthesized quinoline-linked COFs are effective and recyclable photocatalysts for photocatalytic oxidation reactions under harsh conditions, including the organic acid involving oxidative decarboxylation, organic base involving benzylamine coupling, and oxidant involving photocatalytic production of

H₂O₂. This study has demonstrated that enhancing the linkage robustness of photoactive COFs should be a promising strategy to construct heterogeneous photocatalysts for catalytic reactions under harsh conditions.

## Methods
### Synthesis of TAPA-TAPT-COF-Q

A mixture of tris(4-formylphenyl)amine (TFPA) (0.0319 g, 0.096 mmol), 1,3,5-tris-(4-aminophenyl)triazine (TAPT) (0.0354 g, 0.096 mmol), chloranil (0.0077 g, 0.032 mmol) and phenylacetylene (57.1 μL, 0.520 mmol) were added into a 5 mL plastic tube. After adding 1,4-dioxane (1.2 mL) and mesitylene (0.8 ml), the mixture was treated by ultrasonication for 10 min. Then, the mixture was transferred into a 10 mL glass tube. After adding acetic acid (6 M, 0.2 mL) and BF₃•OEt₂ (4.2 μL, 0.032 mmol), ultrasonication for 2 min, and then freezing by liquid nitrogen, it was sealed by three freeze-pump-thaw cycles under vacuum. After cooling down to room temperature, it was put into an oven and reacted at 120 °C for 3 days. The precipitate was collected by filtration, quenched by water (3 × 2 mL), washed by DMF (3 × 2 mL) and THF (3 × 2 mL), and treated by Soxhlet extraction via THF for 24 h. After drying at 60 °C under vacuum for 3 h, the dark brown powder was obtained. Yield: 78.2% (89 mg).

### Synthesis of TAPA-TAPT-COF-Q'

A mixture of TAPA-TAPT-COF (8 mg), chloranil (0.0148 g, 0.064 mmol) and phenylacetylene (10 μL, 0.091 mmol) in toluene (2 mL) were added into a 10 mL glass tube. The mixture was treated by ultrasonication for 10 min to form a homogeneous dispersion. After BF₃•OEt₂ (8.4 μL, 0.064 mmol) was added slowly into the tube, ultrasonication was used for further dispersion. Then, the tube was frozen by liquid nitrogen. It was sealed by three freeze-pump-thaw cycles under vacuum. After cooling down to room temperature, it was put into an oven and

reacted at 120 °C for 3 days. The precipitate was collected by filtration, quenched by water (3 × 2 mL), washed by DMF (3 × 2 mL) and THF (3 × 2 mL), and treated by Soxhlet extraction via THF for 24 h. After drying at 60 °C under vacuum for 3 h, the dark brown was obtained. Yield: 50% (5 mg).

## Gram scale synthesis of TAPA-TAPT-COF-Q

A mixture of tris(4-formylphenyl)amine (TFPA) (0.638 g, 1.92 mmol), 1,3,5-tris-(4-aminophenyl)triazine (TAPT) (0.708 g, 1.92 mmol), chloranil (0.154 g, 0.64 mmol) and phenylacetylene (1.14 mL, 10.40 mmol) were added into a 50 mL reaction flask under $N_2$ atmosphere. After adding 1,4-dioxane (24 mL) and mesitylene (16 ml), the mixture was treated by ultrasonication for 10 min. Then, the mixture was pumped for three times. After adding acetic acid (6 M, 4 mL) and $BF_3 \cdot OEt_2$ (84 μL, 0.64 mmol), it was put into an oil bath and reacted at 120 °C for 3 days. The precipitate was collected by filtration, quenched by water (3 × 2 mL), washed by DMF (3 × 2 mL) and THF (3 × 2 mL), and treated by Soxhlet extraction via THF for 24 h. After drying at 60 °C under vacuum for 3 h, the dark brown powder was obtained. Yield: 54.7% (1.26 g).

## Synthesis of TFPA-TAPT-COF

TFPA-TAPT-COF was synthesized according to the reported method with slight modifications[46,47]. A mixture of tris(4-formylphenyl)amine (TFPA) (0.0319 g, 0.096 mmol) and 1,3,5-tris-(4-aminophenyl)triazine (TAPT) (0.0354 g, 0.096 mmol) were added into a 5 mL plastic tube. After adding 1,4-dioxane (1.2 mL) and mesitylene (0.8 ml), the mixture was treated by ultrasonication for 10 min. Then, the mixture was transferred into a 10 mL glass tube. After adding acetic acid (6 M, 0.2 mL), ultrasonication for 2 min, and then freezing by liquid nitrogen, it was sealed by three freeze-pump-thaw cycles under vacuum. After cooling down to room temperature, it was put into an oven and reacted at 120 °C for 3 days. The precipitate was collected by filtration, washed by DMF (3 × 2 mL) and THF (3 × 2 mL), and treated by Soxhlet extraction via THF for 24 h. After drying at 60 °C under vacuum for 3 h, the orange powder was obtained. Yield: 93.4% (58 mg).

## Synthesis of TAPA-TPB-COF-Q

A mixture of tris(4-formylphenyl)amine (TFPA) (0.0319 g, 0.096 mmol), 1,3,5-tris(4-aminophenyl)benzene (TPB) (0.0340 g, 0.096 mmol), chloranil (0.0077 g, 0.032 mmol) and phenylacetylene (57.1 μL, 0.520 mmol) were added into a 5 mL plastic tube. After adding 1,4-dioxane (1 mL) and mesitylene (1 ml), the mixture was treated by ultrasonication for 10 min. Then, the mixture was transferred into a 10 mL glass tube. After adding acetic acid (6 M, 0.2 mL) and $BF_3 \cdot OEt_2$ (4.2 μL, 0.032 mmol), ultrasonication for 2 min, and then freezing by liquid nitrogen, it was sealed by three freeze-pump-thaw cycles under vacuum. After cooling down to room temperature, it was put into an oven and reacted at 120 °C for 3 days. The precipitate was collected by filtration, quenched by water (3 × 2 mL), washed by DMF (3 × 2 mL) and THF (3 × 2 mL), and treated by Soxhlet extraction via THF for 24 h. After drying at 60 °C under vacuum for 3 h, the brown powder was obtained. Yield: 66% (75 mg).

## Synthesis of TAPA-TPB-COF-Q'

A mixture of TAPA-TPB-COF (8 mg), chloranil (0.0148 g, 0.064 mmol) and phenylacetylene (10 μL, 0.091 mmol) in toluene (2 mL) were added into a 10 mL glass tube. The mixture was treated by ultrasonication for 10 min to form a homogeneous dispersion. After $BF_3 \cdot OEt_2$ (8.4 μL, 0.064 mmol) was added slowly into the tube, ultrasonication was used for further dispersion. Then, the tube was frozen by liquid nitrogen. It was sealed by three freeze-pump-thaw cycles under vacuum. After cooling down to room temperature, it was put into an oven and reacted at 120 °C for 3 days. The precipitate was collected by filtration, quenched by water (3 × 2 mL), washed by DMF (3 × 2 mL) and THF

(3 × 2 mL), and treated by Soxhlet extraction via THF for 24 h. After drying at 60 °C under vacuum for 3 h, the brown powder was obtained. Yield: 40% (4 mg).

## Synthesis of TFPA-TPB-COF

TFPA-TPB-COF was synthesized according to the reported method with slight modifications[46,47]. A mixture of tris(4-formylphenyl)amine (TFPA) (0.0319 g, 0.096 mmol) and 1,3,5-tris(4-aminophenyl)benzene (TPB) (0.0340 g, 0.096 mmol) were added into a 5 mL plastic tube. After adding 1,4-dioxane (1 mL) and mesitylene (1 ml), the mixture was treated by ultrasonication for 10 min. Then, the mixture was transferred into a 10 mL glass tube. After adding acetic acid (6 M, 0.2 mL), ultrasonication for 2 min, and then freezing by liquid nitrogen, it was sealed by three freeze-pump-thaw cycles under vacuum. After cooling down to room temperature, it was put into an oven and reacted at 120 °C for 3 days. The precipitate was collected by filtration, washed by DMF (3 × 2 mL) and THF (3 × 2 mL), and treated by Soxhlet extraction via THF for 24 h. After drying at 60 °C under vacuum for 3 h, the bright yellow powder was obtained. Yield: 74.1% (45 mg).

The crystalline structures of all four COFs were built and refined using the software Materials Studio and the structure coordinates in the AA stacking models are listed in the Supplementary Tables 8–11.

## Data availability

The authors declare that all the data supporting the findings of this study are available within the article. The Supplementary Information, Source Data, and full image dataset are also available from the corresponding author upon request.

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

## Acknowledgements

This work was supported by the "Qilu Young Talent Scholar" program (11190088963032) of Shandong University, the Natural Science Foundation of Shandong Province (ZR202211090168) and the Ministry of Education Singapore under Its Academic Research Funds (RG85/22 and MOET2EP10120-0003). We thank the Analytical Center for Structural Constituent and Physical Property of Core Facilities Sharing Platform, Shandong University, for the structure analyses.

## Author contributions

Jia-Rui Wang data curation, formal analysis, investigation, methodology, writing-original draft; Kepeng Song investigation; Tian-Xiang Luan investigation, methodology; Ke Cheng investigation, methodology; Qiurong Wang investigation, methodology; Yue Wang investigation, methodology; William W. Yu, investigation; Pei-Zhou Li conceptualization, funding acquisition, project administration, writing-review & editing, Yanli Zhao funding acquisition, project administration, writing-review & editing.

## Competing interests

The authors declare no competing interests.
