## [Peer review file · Nature Communications]

REVIEWER COMMENTS

Reviewer #1 (Remarks to the Author):

In this manuscript, the authors reported two robust photoresponsive COFs by modifying imine linkage into quinoline linkage for photocatalytic reactions in harsh conditions. The synthesis methods and structure determinations of the COFs were well validated. Their stability and photocatalytic activity were carefully investigated, especially in harsh conditions. They have demonstrated that the stability and physicochemical properties of COFs could be successfully improved by proper modifications. Therefore, I recommend its publication after minor revisions as shown below:

1. There are various modification methods to enhance the robustness of the imine linkage in COF materials. Why did the authors choose the Povarov reaction as the modification method? Does phenyl group as a side chain in this reaction have any benefit for the photocatalysis reactions?
2. In the article, the authors mentioned that the two synthesis methods (one-pot method and post-modification method) can enhance the robustness of synthesized COFs. Are there any differences between these two methods? Why did the authors take the COF samples synthesized by the one-pot method as the photocatalysts rather than using the COFs synthesized by the post-modification method?
3. As shown in the PXRD characterization, there are M-shaped diffraction peaks in imine-linkage COFs, while it is absent in quinoline-linked COFs. Did these peaks have any influence on the photocatalytic results?
4. As shown in Table 2, the conversion rate and selectivity in pure oxygen environment are higher than that in air. Why did the authors choose air environment as the final reaction condition? More explanations are needed.

Reviewer #2 (Remarks to the Author):

The manuscript reports the synthesis of quinoline-linked COFs and their application for photocatalytic conversion of organic small molecules. The introduction of quinoline linkage improve the stability of COF and some catalytic reaction requiring harsh condition, such as those involving organic acids or bases.

1. The introduction of quinoline linkage is an important linkage conversion chemistry for COF materials, and has been well-demonstrated, such as reported in refs 31-36, and many more. The synthetic methods, both one-pot and post-modification, has been reported before.
2. The aerobic oxidation of sulfide to sulfoxide and the photocatalytic coupling of amine are typical model catalytic reaction and have been demonstrated by a number of different COF with imine, hydrozone, sp² C=C linkages. In particular, the oxidation of sulfide has been demonstrated in Ref. 36 using quinoline COF.

Overall, the synthetic part of this work is lack of novelty. The demonstration of catalytic application in harsh condition is welcome to know. However, the reactions are not so difficult and the advantage of this specific system is elusive. This is an incremental work and doesn't meet the high level for Nat. Commun.

Other minor issues

- 3 In the XPS data shown in Figure 3c and 3f, both N peaks shift to lower binding energy. The sp³ N should not be affected by the introduction of quinoline linkage. Please comment.
- 4 The shift of binding energy of N1s is observed in Figure 6d. However, it is stated that "xxxx were still maintained."
- 5 The values of E_g in the 3rd and 4th materials in Figure 5g are wrong.
- 6 The original report about the construction of quinoline COF is missed (Nat. Commun. 2018)

Reviewer #3 (Remarks to the Author):

This manuscript reported the fabrication of two photoactive quinoline-linked COFs by enhancing linkage robustness for efficient photocatalytic reactions under harsh conditions. The detailed characterizations revealed that quinoline linkages in the COFs not only enhanced the framework stability, but also increased their photochemical properties compared to the pristine imine-linked COFs. This manuscript was well written and the author also gave an in-depth explanation for the outstanding photocatalytic properties, which guarantees its publication in nature communications. Several issues should be addressed prior acceptance.

1. Figure 2c and 2f, the resolution of both pictures should be improved and the benzene rings of the structures look like cyclohexane rings.
2. The authors studied the generation of superoxide radicals of the obtained COFs, how about the singlet oxygen of both COFs. Generally, the photocatalytic oxidation of sulfides is more commonly related to the energy transfer pathway.
3. The format of some references should be unified. For example, ref.1 includes all the authors names, while ref.2 only has the first author. And, the title of each reference should be capital or not, please unify all the references.
4. The first work about transforming imine in COFs into quinoline linkages need to be properly referred. (Nat. Commun., 2018, 9:2998.)

Point-by-point response to reviewers' comments

Reviewer #1:

Comment: "In this manuscript, the authors reported two robust photoresponsive COFs by modifying imine linkage into quinoline linkage for photocatalytic reactions in harsh conditions. The synthesis methods and structure determinations of the COFs were well validated. Their stability and photocatalytic activity were carefully investigated, especially in harsh conditions. They have demonstrated that the stability and physicochemical properties of COFs could be successfully improved by proper modifications. Therefore, I recommend its publication after minor revisions as shown below:"

Answer: We express our sincere thanks to the reviewer for his/her valuable comments and recommendation for the publication after an appropriate revision. Please find our point-by-point responses below.

(1) Comment: "There are various modification methods to enhance the robustness of the imine linkage in COF materials. Why did the authors choose the Povarov reaction as the modification method? Does phenyl group as a side chain in this reaction have any benefit for the photocatalysis reactions?"

Answer: Many thanks for the reviewer's valuable comments. The Povarov reaction not only converts the imine linkages into quinoline linkage to enhance the bond robustness, but also increases the conjugation system by converting the original donor-acceptor system to donor- π -acceptor system, which can extend the photo-absorption capability and finally improve their performance in photocatalysis. Moreover, the Povarov reaction can introduce benzene rings as the side group to enhance the interactions between two neighbor layers via the longitudinal π - π stacking effect to give the generation of a highly crystalline framework. Considering these benefits it possesses, we thus chose the Povarov reaction as the modification method. In our revised manuscript, we have added these discussions to explain it further (page 6).

(2) Comment: "In the article, the authors mentioned that the two synthesis methods (one-pot method and post-modification method) can enhance the robustness of synthesized COFs. Are there any differences between these two methods? Why did the authors take the COF samples synthesized by the one-pot method as the photocatalysts rather than using the COFs synthesized by the post-modification method?"

Answer: Many thanks for the reviewer's valuable comments. Both methods (one-pot method and post-modification method) can synthesize the quinoline linked COFs by transform imine linkages into quinoline linkages as detected by PXRD measurements. The quinoline linked COFs synthesized by the one-pot method had higher yields than the COFs synthesized by the post-modification method. After scaling up experiments by one-pot method, we found that it can even achieve gram scale production. Thus, we take the COF samples synthesized by the one-pot method as the photocatalysts. In our revised manuscript,

we have added the yields of the COFs synthesized by both methods and the scaling up synthesis process with one-pot method in the Methods section (page 34-37). Meanwhile, we have also added the related descriptions of “It should be mentioned that the yield of TFPA-TAPT-COF-Q synthesized by the one-pot method is much higher than that of TFPA-TAPT-COF-Q’ synthesized by the post-modification method. Moreover, TFPA-TAPT-COF-Q could be easily scaled up to the gram level by the one-pot method (see method section). Therefore, TFPA-TAPT-COF-Q synthesized by the one-pot method is used for the following structural analyses and catalytic performance characterizations” (page 8-9).

(3) Comment: “As shown in the PXRD characterization, there are M-shaped diffraction peaks in imine-linkage COFs, while it is absent in quinoline-linked COFs. Did these peaks have any influence on the photocatalytic results?”

Answer: Many thanks for the reviewer’s valuable comments. The M-shaped peaks in the PXRD pattern of imide-linked COFs are usually caused by a stepwise slide of the 2D frameworks in their vertical direction (Ref. 56-58). The stepwise slide between layers usually leads to a decrease of their crystallinity and surface areas of the imide-linked COFs. After transforming the imide-linkages into quinoline-linkages in the COFs, the longitudinal π - π stacking interactions increase, making the framework arrangement more regular and finally giving rise to the result of that the M-shaped peaks disappeared in the PXRD of quinoline-linked COFs. Therefore, it was speculated that the quinoline linkage as well as the side benzene-ring introduced during modification could enhance their longitudinal π - π stacking interactions and the crystallinity of quinoline-linked COFs. We have added the related descriptions of “It is worth mentioning that, in their PXRD patterns, M-shaped diffraction peaks in both imine-linked COFs at the low angle usually caused by a stepwise slide of the 2D frameworks in their vertical direction disappeared, while single sharp peaks were instead observed at the related 2θ areas in these two quinoline-linked COFs. Therefore, it was speculated that the quinoline linkage as well as the introduced side benzene-ring can enhance their longitudinal π - π stacking interactions and the crystallinity of the formed COFs” (page 10).

(4) Comment: “As shown in Table 2, the conversion rate and selectivity in pure oxygen environment are higher than that in air. Why did the authors choose air environment as the final reaction condition? More explanations are needed.”

Answer: Many thanks for the reviewer’s valuable comments. As shown in Table 2, the conversion rate and selectivity under the air environment did not have significant differences from those under the pure oxygen environment. In terms of the real reaction atmosphere, air is cheaper and easier to obtain, and therefore, we finally chose air instead of pure oxygen as the final reaction condition. When investigating the reaction mechanism, in order to eliminate the interferences from other factors such as water in the air, we used the pure oxygen instead and the product becomes only aldehyde. We have added the related descriptions of “When we changed the air to pure oxygen as the gas atmosphere in the

absence of water, the product becomes only aldehyde, demonstrating that the water molecule from the air or solvents in the reaction system is the main reason to lead to the generation of alcohol product" (page 26).

Reviewer #2:

Comment: “The manuscript reports the synthesis of quinoline-linked COFs and their application for photocatalytic conversion of organic small molecules. The introduction of quinoline linkage improve the stability of COF and some catalytic reaction requiring harsh condition, such as those involving organic acids or bases.

1. The introduction of quinoline linkage is an important linkage conversion chemistry for COF materials, and has been well-demonstrated, such as reported in refs 31-36, and many more. The synthetic methods, both one-pot and post-modification, has been reported before.

2. The aerobic oxidation of sulfide to sulfoxide and the photocatalytic coupling of amine are typical model catalytic reaction and have been demonstrated by a number of different COF with imine, hydrozone, sp² C=C linkages. In particular, the oxidation of sulfide has been demonstrated in Ref. 36 using quinoline COF.

Overall, the synthetic part of this work is lack of novelty. The demonstration of catalytic application in harsh condition is welcome to know. However, the reactions are not so difficult and the advantage of this specific system is elusive. This is an incremental work and doesn't meet the high level for Nat. Commun.”

Answer: We express our sincere thanks to the reviewer for his/her valuable comments. Developing heterogeneous photocatalysts for the applications in harsh conditions is of significantly important. Although various photocatalysts have been synthesized for photocatalytic aerobic oxidations so far, developing effective photocatalysts for carrying out aerobic oxidations especially in harsh conditions is still quit challenging. Covalent organic frameworks (COFs) have emerged as a group of functional porous materials in wide applications. Their organic feature endows them with chemical tunability and functionality. Therefore, enhancing the linkage robustness of photoactive COFs becomes a feasible way to the construction of heterogeneous photocatalysts for the applications in harsh conditions. Herein, converting the imine linkages into quinoline groups in triphenylamine incorporated COFs, two photosensitive COFs have been successfully constructed. Our studies revealed that both of their stabilities and photochemical properties are highly enhanced. The high photocatalytic activity combining with their high stability make them suitable for photocatalytic oxidations under normal conditions for sulfide oxidation, and also for recyclable photocatalytic oxidations under relatively harsh conditions such as organic acid involving oxidative decarboxylation and organic base involving benzylamine coupling, demonstrating that enhancing the linkage robustness of photoactive COFs should be a promising strategy to construct heterogeneous catalysts for photocatalytic reactions under harsh conditions. As an important development on enhancing the robustness of photosensitizing COFs for photocatalytic reactions under harsh conditions, we sincerely hope that the revised manuscript is now suitable for publication in the journal.

In the light of the reviewer's comments, we have substantially revised our manuscript. Especially in the part of photocatalytic reactions in harsh conditions, in addition to the photocatalytic reactions involving organic acids or bases, in our revised manuscript, we have

also conducted the photocatalytic reactions under strong oxidizing condition for photocatalytic production of H₂O₂, which is a useful and challenging reaction in heterogeneous photocatalysis. The experiments revealed that the quinoline-linked COFs displayed high efficiency up to 11831.6 μmol·g⁻¹·h⁻¹ and long-term recyclable usability for photocatalytic production of H₂O₂, which is among the highest efficiency using COF-based photocatalysts reported so far for the photocatalytic production of H₂O₂. We have added the results on the photocatalytic H₂O₂ production experiments in **Figure 8** and provided a deep discussion as follows (**pages 31-33**):

“The effective production of H₂O₂ from water under air atmosphere by heterogeneous photocatalysis is a green method in chemical industry due to the safety risks during the long-term storage and transportation of H₂O₂.^{48,51,78-84} Because the generated H₂O₂ is a strong oxidant, the photocatalysts usually should be robust enough in such a strong oxidation condition. Inspired by the high stability in aforementioned strong oxidizing condition and high photocatalytic activity in both photocatalytic reactions of organic acid involving oxidative decarboxylation and organic base involving benzylamine coupling, we then investigated the photocatalytic properties of the prepared quinoline-linked COFs under such a strong oxidation condition for photocatalytic production of H₂O₂. The quinoline-linked TFPA-TAPT-COF-Q was first chosen as a photocatalyst to explore the optimal reaction conditions. As shown in Figure 8a, there was no H₂O₂ production detected under illumination in the absence of air, while an appropriate amount of H₂O₂ was generated in the presence of air, indicating that oxygen in air is the oxygen source for producing H₂O₂. When a variety of alcohols such as ethanol (EtOH), isopropanol (IPA), and triethanolamine (TEOA) were added into the aqueous solution as hole sacrificial reagents, obvious increases in the H₂O₂ production yields were observed. When changing the hole sacrificial reagents from the tested aliphatic alcohols to benzyl alcohol (BA), a further increase in H₂O₂ production yield up to 11831.6 μmol·g⁻¹·h⁻¹ within the first 1 h was observed (Figure 8b). At the same condition, a value of 9251.0 μmol·g⁻¹·h⁻¹ for H₂O₂ production within 1 h was also obtained using TFPA-TPB-COF-Q as the photocatalyst and BA as hole sacrificial reagent (Figure 8b), making these quinoline-linked COFs among the best COF-based photocatalysts for photocatalytic production of H₂O₂ (Supplementary Table 5).^{48,51,78-84} Comparing with other alcohols, higher efficiency of BA should be attributed to the strong π-π stacking interaction between the aromatic region of BA and benzene side chain in the substituted quinoline linkages of the COFs, thus facilitating the electrons transfer to the surface of quinoline-linkage COFs for achieving a higher hole sacrifice rate. For both of the imine-linked TFPA-TAPT-COF and TFPA-TPB-COF, remarkable decreases in the photocatalytic activity for the production of H₂O₂ under the same conditions were observed as shown in Figure 8b. By considering the differences of their structures before and after the modification, the main reason should be that when converting imine linkages into quinoline linkages, the original donor-acceptor system in imine-linked COFs was changed into donor-π-acceptor system, which can increase the conjugation, extend the photo-absorption capability and finally improve the performance in photocatalysis. When compared with other reported COF-based

photocatalysts possessing similar structures (Supplementary Table 5), higher performance of both TFPA-TAPT-COF-Q and TFPA-TPB-COF-Q should also be attributed to the introduction of the photoactive TPA moieties into the COF skeletons.

Motivated by the high efficiency of the quinoline-linkage COFs in photocatalytic production of H₂O₂, their long-term photocatalytic capacity and recyclability were carried out to verify the robustness against the strong oxidizing environment (Figure 8c,d). As shown in Figure 8c, under continuous illumination, the H₂O₂ production yields with TFPA-TAPT-COF-Q and TFPA-TPB-COF-Q as photocatalysts can increase steadily up to 295900 μmol·g⁻¹ and 288090 μmol·g⁻¹ over time respectively, while the pristine imine linked COFs are easily decomposed to leave only a small amount of solids under the same conditions after about 24 h continuous reactions. Moreover, the photocatalytic activity of TFPA-TAPT-COF-Q and TFPA-TPB-COF-Q was not decreased even after 5 cycles of reactions based on the detection results of recollected samples for every 4 h per time (Figure 8d). The PXRD measurements of the recollected samples after 24 h reactions also confirmed that the crystallinity of the quinoline-linked TFPA-TAPT-COF-Q and TFPA-TPB-COF-Q was still maintained, while main peaks in the PXRD patterns of the imine-linked COFs completely disappeared after 12 h reactions (Supplementary Figures 57-59).

Finally, the mechanism for photocatalytic H₂O₂ production was investigated by quenching experiments (Figure 8a). When tert-butyl alcohol (TBA) replacing the sacrificial agent of BA as the trapping agent for hydroxyl radical (•OH) was added to the mixture, the H₂O₂ production yield was found nearly the same as the case in the absence of the sacrificial agent. When KI or BQ as the typical trapping agent for O₂•⁻ was added to the mixture solvent of water and BA, the yields decreased sharply or even no H₂O₂ production was detected. These results suggest that the photocatalytic formation of O₂•⁻ by O₂ binding with electrons plays a key role in the reaction rather than •OH. Meanwhile, the benzaldehyde and benzoic acid were also detected in the liquid phase, indicating that BA was oxidized by photogenerated holes (Figure 8e). Finally, the O₂•⁻ radical combines with the lost protons to give the formation of H₂O₂ (Supplementary Figure 60).^{48,51,78-84}

We have added the stability measurements for the quinoline-linkage COFs in strong oxidization condition as described in the sentences of “Subsequently, the chemical stability of the quinoline-linked samples after immersion in aqueous solutions with different concentrations of acids and bases and alcohol solutions containing oxidizing and reducing agents was studied by the PXRD measurements. No matter immersed in 3 and 6 mol/L of HCl or NaOH solutions, ethanol solution containing H₂O₂, or methanol solution containing NaBH₄ for 12 h, the main peaks of the PXRD patterns for TFPA-TAPT-COF-Q and TFPA-TPB-COF-Q were still maintained well (Figure 4c,d). For TFPA-TAPT-COF and TFPA-TPB-COF, although the protonation phenomena as previously reported⁵⁶⁻⁵⁸ was observed in relatively low concentration of acid (Supplementary Figures 23-25), a remarkable decrease of the main peak in their PXRD patterns was detected after long-term soaking in a higher concentration of acid, and the main peaks in their PXRD patterns sharply decreased after immersing in

either low or high concentration of base solutions. The same results were also observed in their PXRD patterns of imine-linked COFs after immersing in alcohol solutions containing oxidizing and reducing agents (Figure 4e,f). The stability investigations revealed that after converting the imine linkages into quinoline linkages, TFPA-TAPT-COF-Q and TFPA-TPB-COF-Q can maintain their chemical stability well in harsh acid, base and oxidizing and reducing agent solutions.” (page 15).

Considering photocatalytic oxidation of sulfides is a type reaction under normal condition instead of harsh condition, we have moved the main results of the photocatalytic oxidation of sulfides into the Supplementary Information (Tables S2-S4 and Figures S30-S44) and shortened the discussions about the photocatalytic oxidation of sulfides as shown in the sentences of “Photocatalytic oxidation of sulfide as a typical reaction with the process by photocatalytically generating $O_2^{\bullet-}$ is often taken as a model reaction to detect the activity of the photocatalysts.^{19-22,71,72} By taking TFPA-TAPT-COF-Q as the catalyst, we first investigated its photocatalytic activity on aerobic oxidation of sulfide with methylphenyl sulfide as the representative substrate (Table S2). Then, we also measured other synthesized COFs under the same optimal conditions. The studies revealed that all of the four synthetic COFs displayed good recyclable photocatalytic performance (Supplementary Figures 30-43). In addition, their high photocatalytic performance still possessed when extending the substrates to other sulfides (Supplementary Tables 3 and 4) upon photocatalytically generating $O_2^{\bullet-}$ (Supplementary Figure 44).” (page 20-21).

We have also changed the discussions in the Abstract and Discussion as shown in the sentences of “The high photocatalytic activity combining with their good stability make them suitable photocatalysts for photocatalytic reactions under harsh conditions, as successfully verified by the recyclable photocatalytic reactions of organic acid involving oxidative decarboxylation and organic base involving benzylamine coupling. Under strong oxidative condition, the quinoline-linked COFs display a high efficiency up to $11831.6 \mu\text{mol}\cdot\text{g}^{-1}\cdot\text{h}^{-1}$ and a long-term recyclable usability for photocatalytic production of H_2O_2 , while the pristine imine-linked COFs are less catalytically active and easily decomposed in these harsh conditions.” (page 1-2), and in the sentences of “Photocatalytic experiments confirm that the synthesized quinoline-linked COFs are effective and recyclable photocatalysts for photocatalytic oxidation reactions under harsh conditions, including the organic acid involving oxidative decarboxylation, organic base involving benzylamine coupling, and oxidant involving photocatalytic production of H_2O_2 .” (page 34).

In addition, we have carefully revised the manuscript according to the comments, and please find our point-by-point responses below.

(1) Comment: “In the XPS data shown in Figure 3c and 3f, both N peaks shift to lower binding energy. The sp^3 N should not be affected by the introduction of quinoline linkage. Please comment.”

Answer: Many thanks for the reviewer's valuable comments. In our revised manuscript, we have identified the original XPS peaks once again. The peak at ~ 400.1 eV was classified as sp^3 -N in triphenylamine in the imine-linked COFs based on the references 38-51. The signal of sp^3 -N at ~ 400.1 eV was slightly shift to ~ 399.9 eV after modification. To explain this peak shift, we also did XPS measurements for the powder samples of the monomers. The results of fractional peak fitting showed that the XPS peak of the original TFPA monomer was at 400.31 eV. After the formation of COFs, the peak position slightly shifted to lower binding energy at ~ 400.1 eV in the imine-linked COFs and ~ 399.9 eV in quinoline linked COFs. Thus, we think that when the TFPA monomer forms COFs, the electron distribution of N atoms is broadened due to the formation of the donor-acceptor system in the imine-linked COFs and donor- π -acceptor system in quinoline linked COFs, finally making the peak position shifted to lower binding energy. We have added the results and discussions of the peak shifts in the sentences of "X-ray photoelectron spectroscopy (XPS) measurements were also carried out to verify the chemical states of linked N atoms in these COFs. The peak of the nitrogen in the -C=N- imine fragment and triazine moieties at ~ 398.7 eV in TFPA-TAPT-COF shifted to lower binding energy at ~ 398.6 eV with a broadened full width at half maxima (FWHM) after modification, indicating the success conversion of imine linkages to quinoline linkages. Moreover, XPS spectra showed that the signals of N 1s in the triphenylamine fragments at 400.31 eV in the original TFPA monomer slightly shifted to ~ 400.1 eV in the imine-linked COFs and ~ 399.9 eV after forming quinoline linked COFs, which indicate that after the synthesis from TFPA monomer to COFs, the electron distribution of N atoms is broadened due to the formation of their donor-acceptor system in TFPA-TAPT-COF and donor- π -acceptor system in the quinoline linked TFPA-TAPT-COF-Q lowering the binding energy of N 1s in the triphenylamine fragments" (page 12). The XPS measurement results of the monomers were also added as Figures S11 and S12 in the Supplementary Information.

(2) Comment: "The shift of binding energy of N1s is observed in Figure 6d. However, it is stated that "xxxx were still maintained.""

Answer: Many thanks for the reviewer's valuable comments. We re-calibrated the peaks in XPS measurement results with the carbon peak at 284.8 eV as the reference peak. The peak shift of N1s before and after photocatalysis based on the result of re-calibration is only 0.02eV, which should be a reasonable error caused by the instrument or the deconvolution. Considering that photocatalytic oxidation of sulfides is a type reaction under normal condition instead of harsh condition, we have moved the main results of the photocatalytic oxidation of sulfides into the Supplementary Information (Tables S2-S4 and Figures S30-S44) and shortened the discussions about the photocatalytic oxidation of sulfides as shown in the sentences of "Photocatalytic oxidation of sulfide as a typical reaction with the process by photocatalytically generating $O_2^{\bullet-}$ is often taken as a model reaction to detect the activity of the photocatalysts.^{19-22,71,72} By taking TFPA-TAPT-COF-Q as the catalyst, we first investigated its photocatalytic activity on aerobic oxidation of sulfide with methylphenyl sulfide as the representative substrate (Table S2). Then, we also measured other synthesized COFs under

the same optimal conditions. The studies revealed that all of the four synthetic COFs displayed good recyclable photocatalytic performance (Supplementary Figures 30-43). In addition, their high photocatalytic performance still possessed when extending the substrates to other sulfides (Supplementary Tables 3 and 4) upon photocatalytically generating $O_2^{\bullet-}$ (Supplementary Figure 44)." (page 20-21).

(3) Comment: "The values of E_g in the 3rd and 4th materials in Figure 5g are wrong."

Answer: Many thanks for the reviewer's valuable comments. In our revised manuscript, we have corrected the E_g values of TFPA-TPB-COF-Q and TFPA-TPB-COF to 1.80 eV and 2.42 eV, respectively (Figure 5g).

(4) Comment: "The original report about the construction of quinoline COF is missed (Nat. Commun. 2018)"

Answer: Many thanks for the reviewer's valuable comments. In our revised manuscript, we have cited the original report about the construction of quinoline COFs as ref. 38.

Reviewer #3:

Comment: “This manuscript reported the fabrication of two photoactive quinoline-linked COFs by enhancing linkage robustness for efficient photocatalytic reactions under harsh conditions. The detailed characterizations revealed that quinoline linkages in the COFs not only enhanced the framework stability, but also increased their photochemical properties compared to the pristine imine-linked COFs. This manuscript was well written and the author also gave an in-depth explanation for the outstanding photocatalytic properties, which guarantees its publication in nature communications. Several issues should be addressed prior acceptance.”

Answer: We express our sincere thanks to the reviewer for his/her valuable comments and recommendation for the publication after an appropriate revision. Please find our point-by-point responses below.

(1) Comment: “Figure 2c and 2f, the resolution of both pictures should be improved and the benzene rings of the structures look like cyclohexane rings.”

Answer: Many thanks for the reviewer’s valuable comments. In our revised manuscript, we have improved the resolution of the figures, especially Figure 2c and 2f. Moreover, in order to avoid potential misunderstanding, we have also emphasized the model types in the caption as described in "Perspective view (left) and side view (right) of TFPA-TAPT-COF-Q in AA stacking ball-and-stick model (H atoms are omitted for clarity)" for **Figure 2c** and "Perspective view (left) and side view (right) of TFPA-TPB-COF-Q in AA stacking ball-and-stick model (H atoms are omitted for clarity)" for **Figure 2f**.

(2) Comment: “The authors studied the generation of superoxide radicals of the obtained COFs, how about the singlet oxygen of both COFs. Generally, the photocatalytic oxidation of sulfides is more commonly related to the energy transfer pathway.”

Answer: Many thanks for the reviewer’s valuable comments. In the photocatalytic oxidation of sulfides, both superoxide radical and singlet oxygen play an important role. We have conducted the mechanism experiments once again after adding different radical trapping groups, and the results revealed that the sulfide oxidation reactions photocatalyzed by our synthesized quinoline COFs are mainly dominated by superoxide anion radical (Entries 14-16, Table 1). We have performed the EPR measurements of all four COFs, and no singlet oxygen generation was detected under the condition of TEMP as the trapping agent and acetonitrile as the solvent. These results confirmed the role of superoxide radical in photocatalytic oxidation of sulfides. In our revised manuscript, we have added the discussion of “Meanwhile, the same measurements were also applied to detect the production of singlet oxygen ($^1\text{O}_2$) with 2,2,6,6-tetramethylpiperidine (TEMP) as the commonly used spin trapping agent. The EPR measurement results revealed that no signal of $^1\text{O}_2$ was detected no matter under light illumination or not (Figures S28 and S29)” (**page 18-19**) and also the spectra of the EPR measurements as **Figures S28 and S29** in the Supplementary Information.

(3) Comment: “The format of some references should be unified. For example, ref.1 includes all the authors names, while ref.2 only has the first author. And, the title of each reference should be capital or not, please unify all the references.”

Answer: Many thanks for the reviewer’s valuable comments. In our revised manuscript, the format of all the cited references has been unified according to the journal guidelines (papers with more than five authors are listed with the first author followed by “*et al.*”).

(4) Comment: “The first work about transforming imine in COFs into quinoline linkages need to be properly referred. (Nat. Commun., 2018, 9:2998.)”

Answer: Many thanks for the reviewer’s valuable comments. In our revised manuscript, we have added citation of the original report on the construction of quinoline COFs (Nat. Commun. 2018, 9:2998) as **ref. 38**.

REVIEWERS' COMMENTS

Reviewer #1 (Remarks to the Author):

In this revised version, the authors have carefully made the corrections and responses according to the reviewers' comments. I agree with accepting this paper for publication at its current state.

Reviewer #2 (Remarks to the Author):

The revised manuscript has properly addressed the reviewers comments and concerns. It is recommended for publishing.

Reviewer #3 (Remarks to the Author):

The authors had made the necessary experiments and answered all my questions. Therefore, I recommended the acceptance in Nat. Commun. at this stage.

Response to Reviewers' Comments

Reviewer #1 (Remarks to the Author):

In this revised version, the authors have carefully made the corrections and responses according to the reviewers' comments. I agree with accepting this paper for publication at its current state.

Our response: Thanks very much for your recommendation of publication.

Reviewer #2 (Remarks to the Author):

The revised manuscript has properly addressed the reviewers comments and concerns. It is recommended for publishing.

Our response: Thanks very much for your recommendation of publication.

Reviewer #3 (Remarks to the Author):

The authors had made the necessary experiments and answered all my questions. Therefore, I recommended the acceptance in Nat. Commun. at this stage.

Our response: Thanks very much for your recommendation of publication.